# Contextual Influence on Pattern Separation During Encoding

**DOI:** 10.3390/neurosci6010013

**Published:** 2025-02-06

**Authors:** Laura García-Rueda, Claudia Poch, Joaquín Macedo-Pascual, Pablo Campo

**Affiliations:** 1PhD Program in Neuroscience, Autonomous University of Madrid-Cajal Institute, 28029 Madrid, Spain; laura.garciarueda@estudiante.uam.es; 2Facultad de Lenguas y Educación, Universidad de Nebrija, 28015 Madrid, Spain; cpoch@nebrija.es (C.P.); jmacedo@nebrija.es (J.M.-P.); 3Department of Basic Psychology, Autonomous University of Madrid, Campus de Cantoblanco, 28049 Madrid, Spain

**Keywords:** subsequent memory effect, ERPs, episodic memory, mnemonic discrimination, pattern separation, memory–context dependency

## Abstract

Pattern separation is considered a crucial process that allows us to distinguish among the highly similar and overlapping experiences that constitute our episodic memory. Not only do different episodes share common features, but it is often the case that they share the context in which they occurred. While there have been a great number of studies investigating pattern separation and its behavioral counterpart, a process known as mnemonic discrimination, surprisingly, research exploring the influence of context on pattern separation or mnemonic discrimination has been less common. The available evidence shows that similar items with similar contexts led to a failure in pattern separation due to high similarity that triggers overlap between events. On the other hand, others have shown that pattern separation can take place even under these conditions, allowing humans to distinguish between events with similar items and contexts, as different hippocampal subfields would play complementary roles in enabling both pattern separation and pattern completion. In the present study, we were interested in testing how stability in context influenced pattern separation. Despite the fact that pattern separation is by definition an encoding computation, the existing literature has focused on the retrieval phase. Here, we used a subsequent memory paradigm in which we manipulated the similarity of context during the encoding of visual objects selected from diverse categories. Thus, we manipulated the encoded context of each object category (four items within a category), so that some categories had the same context and others had a different context. This approach allowed us to test not only the items presented but also to include the conditions that entail the greatest demand on pattern separation. After a 20 min period, participants performed a visual mnemonic discrimination task in which they had to differentiate between old, similar, and new items by providing one of the three options for each tested item. Similarly to previous studies, we found no interaction between judgments and contexts, and participants were able to discriminate between old and lure items at the behavioral level in both conditions. Moreover, when averaging the ERPs of all the items presented within a category, a significant SME emerged between hits and new misses, but not between hits and old false alarms or similar false alarms. These results suggest that item recognition emerges from the interaction with subsequently encoded information, and not just between item memory strength and retrieval processes.

## 1. Introduction

Episodic memory refers to the ability to form and retrieve memories of specific past events [1]. This simple description, however, hides a highly complex process. Human beings encode an immense amount of life episodes, each of them associated with a spatial and temporal moment [2], that constitute the context. Commonly, context refers to the general information that is associated with a specific event at the time of encoding [3]. The use of this contextual information is a defining feature of episodic memory [1,4,5,6]. Although each event is unique, it is often the case that it shares common features with previous episodes and even shares the context in which they occurred. When we encode similar information, there is a risk of interference between memories due to overlapping neural representations [7,8]. How, then, can humans distinguish among these memories? Our brain uses a process known as pattern separation to form unique representations for similar experiences, allowing us to recollect specific details [9,10,11,12,13,14]. Pattern separation, thus, implies the capacity to resolve interference from an overlap in stimulus features and associated neural responses [15].

In the last decade, there have been a great number of studies investigating the neural correlates of pattern separation, and its behavioral counterpart, a process known as mnemonic discrimination [16,17]. Surprisingly, research exploring the influence of context on pattern separation or mnemonic discrimination has been less common [18,19,20,21,22,23,24,25]. Thus, differentially from previous pattern separation research, these studies used the integration of pictured objects in contexts. Although with certain variations, they commonly presented images of objects in the background of a unique scene at encoding. Interestingly, they showed that background context facilitated target recognition but also increased the rate of false recognition of similar lure items [19,23,24,25], reflecting a reduced mnemonic discrimination. Crucially, Libby, Reagh, Bouffard, Ragland, and Ranganath [22] found that hippocampal activity generalized across similar objects that were encoded in the same context (see also [20]).

In the present study, we were interested in investigating the effect of context on mnemonic discrimination and its neural counterpart (i.e., pattern separation). Several aspects differentiate our study from previous research. First, even though pattern separation is by definition an encoding computation [26], the above-mentioned studies have focused on the retrieval phase. Second, several studies have shown that interference between memory traces is frequently accounted for by their associations with similar contexts [20]. Thus, differentially from previous research, we explored whether successful target identification and mnemonic discrimination can be predicted by ERPs during encoding [21,27,28], when the object–context binding is formed [29]. Finally, previous research has demonstrated that mnemonic discrimination decreased when studying an increased number of related items [30,31]. Thus, visual objects were selected from diverse categories, and, in contrast to previous studies, for each of these categories, four exemplars were presented. With this aim, we used EEG to register the neural activity during a mnemonic discrimination task [10,32,33] and applied a subsequent memory approach.

Since most of the previous studies have used fMRI, very little is known about the temporal dynamics of pattern separation [21,28,33]. Thus, by recording ERPs during our mnemonic discrimination task, we can increase the understanding of the temporal dimension of pattern separation. Furthermore, as recently stated by Amer and Davachi [15], there is an increased interest in the contribution of extra-hippocampal regions to the process of pattern separation, which is best considered a process supported by a network of brain regions [15,34,35,36]. Finally, ERPs are considered a valuable tool to investigate the neural activity at encoding in the subsequent memory paradigm due to the fast and brief nature of neurocognitive processes allowing the separation of different subprocesses [37].

Since we were interested in studying the influence of contextual stability at encoding [38], in this task, participants learned visual objects from a category, which were presented either in the same or different background context. Following previous studies [38,39], in the recognition phase, visual objects were presented without their correspondent background context [23,29] and participants had to differentiate between old (studied object), similar (new items from a previously seen category), and new items (new objects from a new category). Considering the available evidence, we predict that accuracy for target identification will increase for objects presented in the same background context relative to objects presented in different contexts. On the other hand, correct identification of a similar lure (i.e., mnemonic discrimination) will decrease for objects presented in the same background context relative to objects presented in different contexts, because they match both object and context information to a greater degree [30,40]. This can be explained by a generalization process that occurs at the expense of detailed memory for those objects [41]. Regarding the ERP correlates, a recent review [37] proposed a functional organization of ERP SMEs into three main components: two early frontal and parietal components starting at about 300 ms after stimulus onset, reflecting semantic processing and the binding of multiple features of the event, respectively, and a third component, described as a sustained late frontal (starting at around 550 ms) component reflecting associative and conceptual encoding. Previous studies have also suggested that mnemonic operations promoting later memorability can be divided into subprocesses, one starting at 300 ms and the other at around 500 ms [42,43,44]. Accordingly, we hypothesized that correct target identification would be related to a greater positive amplitude starting at around 300 and around 500 ms in the frontal and centroparietal electrodes at the encoding phase. Similar lures incorrectly identified as old objects will exhibit a greater positive amplitude in the frontal and centroparietal electrodes at the encoding phase, similar to the increase in correct recalls [20,21,31]. We also predict that objects encoded in the same context will show greater amplitude than those encoded in different contexts.

## 2. Methods

### 2.1. Participants

Twenty-six students participated in this research. None of the participants reported a medical history of neurological or psychiatric disorders. The group of participants consisted of 20 females and 6 males, with a mean age of 26.77 years (SD = 4.21). Each participant signed an informed consent form, detailing the study procedures in accordance with the 1991 Declaration of Helsinki. All participants completed the encoding and recognition phase of the study under the same conditions. The sample size was calculated using G*Power [45]. With the aim of detecting an effect size of 0.25 and obtaining a statistical power of 0.95, the required sample was determined to be at least 18 participants.

### 2.2. Stimuli and Procedure

The stimuli and procedure used in this study were adapted from the methods described by Poch, Prieto, Hinojosa, and Campo [12]. A total of 1280 images were shown to participants during the study phase. In the instructions provided to participants, they were asked to pay attention to all the images but informed that they would only be asked about the objects that appeared in them. These images represented objects (without human body parts or animals) from different categories, and each category contained four images (see Figure 1). The categories were divided into two groups: same context (160 categories) or different context (160 categories). Each category from the same-context group had four different category objects (a total of 640 images), but all of them had the same background. Each category from the different-context group also had four different category objects (a total of 640 images), but each one of them had a different background. In addition, backgrounds were not repeated across categories, i.e., each category had a unique background (or backgrounds in the case of those belonging to the “different context” group). The presentation of each image lasted for 1500 ms, and a 1000 ms gray screen was displayed after each image. In addition, the images were presented in four blocks of 320 images each, allowing participants to take untimed breaks between blocks. Participants had a 20 min break after completing the study phase and then moved on to the discrimination phase.

In the discrimination phase, 400 images were randomly presented to each participant. In line with the methods by Poch, Prieto, Hinojosa, and Campo [12], we tested the first object presented within each category to ensure that the effects were due to stimulus interference. These images were divided into 160 old items (previously presented in the study phase), 160 lure items (new images, but from a previously seen object category), and 80 new items (new images from new object categories). Furthermore, the old and similar items were divided into two groups: categories encoded with the same context or categories encoded with different contexts. Thus, there were 80 old items from the categories encoded with the same context, 80 old items from the categories encoded with different contexts, 80 lure items from the categories encoded with the same context, 80 lure items from the categories encoded with different contexts, and finally, 80 new items. All images were presented without backgrounds for 1500 ms each and followed by a 2000 ms white screen. Participants were verbally instructed to press key numbers 1, 2, or 3 while the image was on the screen to indicate whether the items were old, lure, or new, respectively. Additionally, these instructions were displayed on the screen before starting the test. During the trials, these response options were not displayed on the screen, but during each break, the instructions were shown on the screen again to remind participants of the correct key–response associations, so that they did not have to rely on memory during the break. Participants were given a non-timed pause at the midpoint of the discrimination phase, after the first 200 images, to reduce fatigue and maintain focus. The break was untimed to allow participants to rest adequately and resume when they felt ready, ensuring that performance in the second half of the task was not compromised by fatigue or time pressure. The test phase lasted approximately 24 min, so it was divided into two blocks of 12 min each. During the encoding phase, the same criterion was followed by dividing the study into blocks of 320 pictures, so each block lasted approximately 13 min. We took into account previous studies that suggest that the performance decline typically occurs within the first 15 min [46,47,48,49,50,51,52].

### 2.3. EEG Acquisition and Processing

Data acquisition for the encoding phase used a Biosemi ActiveTwo system equipped with 128 electrodes, along with extra electrodes for vertical and horizontal electrooculography (EOG) and nose-tip reference. Initial online referencing was conducted via sensors located in the cap’s posterior region (CMS and DRL). The data were digitized at a sampling rate of 2048 Hz and later re-referenced offline to the nose tip. Data processing continued in Matlab, using the Fiedltrip toolbox (version 20201113, accessible at www.fieldtriptoolbox.org, accessed on 28 January 2025), where data were downsampled to 256 Hz. The toolbox operates within Matlab’s R2019b environment (The MathWorks, Natick, MA, USA).

The raw continuous data were divided into epochs of 1700 ms duration, spanning from −500 to 1200 ms around the trial presentation. Following segmentation, an infomax independent components analysis was conducted to remove artifacts related to horizontal eye movements and blinks. Epochs that contained other types of artifacts were discarded based on visual inspection criteria. The signal underwent low-pass filtering with a cutoff frequency of 30 Hz and was averaged for each condition and participant separately. Subsequently, the event-related potentials (ERPs) were baseline corrected using a −200 ms interval before averaging the ERPs across conditions and participants.

### 2.4. Statistical Analyses

#### 2.4.1. Behavioral Data

Analyses were performed using IBM SPSS Statistics 21.0 for Windows. As new items had no context, we performed two different repeated measures analyses of variance (ANOVAs). First, following Stark, Kirwan, and Stark [17], we ran a repeated measures analysis of variance (ANOVA), where a category is a within-subject factor with nine levels (old, similar, and new responses to targets, lures, and foils). Second, we also performed a repeated measures ANOVA where a category is a within-subject factor with six levels (old, similar, and new responses to targets and lures), and context was treated as a two-level (same and different) within-subject factor. Following the ANOVA tests, we applied Bonferroni-corrected pairwise comparisons to identify which means exhibited statistically significant differences. The same analytical procedures were applied for reaction time data, but the old response to the foil condition was excluded in the first analysis due to missing values.

Additionally, we calculated a lure discrimination index (LDI) in terms of context, which was defined as the ability to reject similar lures and was calculated as the proportion of correctly identified lures corrected for the baseline rate of similar responses to novel items [33]. We used a paired samples *t*-test to compare the LDI from the same- and different-context groups.

#### 2.4.2. EEG Data

ERPs were analyzed using a nonparametric cluster-based random permutation analysis approach [53] in the two windows of interest (300–500 ms and 500–800 ms). By using this type of analysis, we were able to identify the spatial distribution of statistical effects while effectively handling the multiple-comparisons problem. In this analysis, permutation tests were employed to calculate the sampling distribution for a cluster-based statistic. Cluster-based statistics involve aggregating spatially and temporally adjacent variables, such as t or F values, into clusters. The definition of the cluster statistic may rely on its maximum value, extent, or a combination of these factors [43]. The analytical procedure followed these steps: First, a parametric statistical test was conducted for each electrode within the specified time window of interest. *p*-values below 0.05 were utilized to identify clusters formed by adjacent electrodes, with a minimum of two channels required for cluster formation. The cluster-level statistic was determined by summing the individual t-statistics or F values within each cluster. Subsequently, a null distribution was generated by calculating 1.000 randomized cluster-level test statistics. The observed cluster-level test statistic was then compared to the null distribution, considering only those clusters that exceeded the 95th percentile as significant.

#### 2.4.3. Encoding Analysis

Based on a previous study [27], three different analyses were carried out: category analysis, first item analysis, and last item analysis, which are described below.

First, in the category analysis, similar presented items within a certain category were averaged as a function of their associated response (see [31]). We used repeated measures ANOVA with two within-subject factors. The response is a factor with five levels: old hit, old false alarm, new miss, similar false alarm, and similar correct rejection. Context is a factor with two levels: the same context and different contexts.

In the first item analysis, we compared the ERPs of a specific item which is the first object presented within a given category. In this case, we used a two-way repeated measures ANOVA with two factors, where the response is a within-subject factor with three levels (old hit, old false alarm, and new miss), and context is a within-subject factor with two levels (same context and different context).

Finally, we analyzed the ERP responses of the last item analysis, which concerned the last studied item within a category (i.e., the fourth presentation), and then averaged as per the recognition judgment of the probed item belonging to that category. We then used a two-way repeated measures ANOVA as in the category analysis.

Therefore, these analyses not only tested the items presented but also included the conditions that entail the greatest demand on pattern separation [19].

## 3. Results

### 3.1. Behavioral Results

The first ANOVA showed a significant main effect of category (F_8,25_ = 79.714, *p* < 0.001, η_p_^2^ = 0.761). As expected, Bonferroni pairwise comparisons showed that new responses to foils (M = 81.183; SD = 2.035) were higher than all other responses (all *p* < 0.001), and old responses to foils (M = 1.887; SD = 0.473) were lower than all other responses (all *p* < 0.001). Old responses to targets (M = 42.900; SD = 2.947) were higher than new responses to targets (M = 20.933; SD = 2.258) (*p* < 0.01), old responses to lures (M = 22.209; SD = 2.245) (*p* < 0.001), and similar responses to foils (M = 16.930; SD = 1.838) (*p* < 0.001). Crucially, similar responses to lures (M = 48.674; SD = 3.004) were higher than old responses to lures (M = 22.209; SD = 2.245) (*p* < 0.001), similar responses to targets (M = 36.167; SD = 3.093) (*p* < 0.001), new responses to targets (M = 20.933; SD = 2.258) (*p* < 0.001), new responses to lures (M = 29.117; SD = 2.692) (*p* < 0.05), and similar responses to foils (M = 16.930; SD = 1.838) (*p* < 0.001). Similar responses to targets (M = 36.167; SD = 3.093) were higher than similar responses to foils (M = 16.930; SD = 1.838) (*p* < 0.001). New responses to lures (M = 29.117; SD = 2.692) were higher than new responses to targets (M = 20.933; SD = 2.258) (*p* < 0.001) (see Figure 2A, see Table 1).

The second ANOVA showed a significant main effect of category (F_5,25_ = 14.104, *p* < 0.001, η_p_^2^ = 0.361). No significant effect for context (F_1,25_ = 1.401, *p* > 0.20, η_p_^2^ = 0.053), nor a significant interaction between category and context (F_5,25_ = 0.436, *p* > 0.75, η_p_^2^ = 0.017) were found. Bonferroni pairwise comparisons showed that similar responses to lures (M = 48.677; SD = 3.003) were higher than old responses to lures (M = 22.197; SD = 2.243) (*p* < 0.001), similar responses to targets (M = 36.176; SD = 3.097) (*p* < 0.001), new responses to targets (M = 20.929; SD = 2.259) (*p* < 0.001), and new responses to lures (M = 29.126; SD = 2.693) (*p* < 0.05). Moreover, old responses to targets (M = 42.895; SD = 2.950) were higher than old responses to lures (M = 22.197; SD = 2.243), and new responses to targets (M = 20.929; SD = 2.259) (all *p* < 0.001). Similar responses to targets (M = 36.176; SD = 3.097) were higher than new responses to targets (M = 20.929; SD = 2.259) (*p* < 0.05). New responses to lures (M = 29.126; SD = 2.693) were higher than new responses to targets (M = 20.929; SD = 2.259) (*p* < 0.001) (see Figure 2B).

The dependent-sample test did not show a significant difference in LDI between the same and different contexts (t(25) = 0.445, *p* > 0.05) (see Figure 3).

### 3.2. Reaction Time Results

The reaction time analysis (see Figure 4 and Table 2) showed a significant main effect of category (F(3.301, 82.523) = 13.805, *p* < 0.01, η_p_^2^ = 0.356) in the first ANOVA. Bonferroni pairwise comparisons showed that similar responses to lures (M = 1.054; SD = 0.079) were slower than old responses to targets (M = 0.962; SD = 0.082) (*p* < 0.01), similar responses to targets (M = 1.030; Sd = 0.086) (*p* < 0.01), new responses to targets (M = 0.955; SD = 0.113) (*p* < 0.001), old responses to lures (M = 0.986; SD = 0.083) (*p* < 0.05), new responses to lures (M = 0.955; SD = 0.114) (*p* < 0.01), and new responses to foils (M = 0.933; SD = 0.099) (*p* < 0.001). In addition, similar responses to foils (M = 1.040; SD = 0.098) were slower than old responses to targets (M = 0.962; SD = 0.082) (*p* < 0.05), new responses to targets (M = 0.955; SD = 0.113) (*p* < 0.001), new responses to lures (M = 0.955; SD = 0.114) (*p* < 0.01), and new responses to foils (M = 0.933; SD = 0.099) (*p* < 0.001). Similar responses to targets (M = 1.030; SD = 0.017) were slower than new responses to targets (M = 0.955; SD = 0.113) (*p* < 0.01), new responses to lures (M = 0.955; SD = 0.114) (*p* < 0.05), and new responses to foils (M = 0.933; SD = 0.099) (*p* > 0.001). The second ANOVA did not show a significant interaction between category and context (F(4.228, 105.695) = 0.400), *p* > 0.05, η_p_^2^ = 0.142), and showed a category main effect (F(0.176, 0.017) = 10.438), *p* < 0.001, η_p_^2^ = 0.996). Bonferroni pairwise comparisons showed that similar responses to lures (M = 1.054; SD = 0.016) were slower than old responses to targets (M = 0.962; SD = 0.016) (*p* < 0.01), old responses to lures (M = 0.986; SD = 0.016) (*p* < 0.05), similar responses to targets (M = 1.030; SD = 0.017), (*p*< 0.01), new responses to targets (M = 0.955; SD = 0.022) (*p* < 0.001), and new responses to lures (M = 0.955; SD = 0.022) (*p* < 0.01). In addition, similar responses to targets were slower than old responses to targets (M = 0.962; SD = 0.016) (*p* < 0.5), new responses to targets (M = 0.955; SD = 0.022), (*p* < 0.01), and new responses to lures (M = 0.955; SD = 0.022) (*p* < 0.01).

### 3.3. ERP Results

#### Encoding

The category analysis showed that ERP amplitudes were significantly different between 500 and 800 ms (*p* < 0.05; d’ = 0.29). This effect was explained by a more positive deflection in a central-anterior cluster of electrodes for the old hits compared to the new miss responses, which was more evident on the right side (*p* < 0.05) (Figure 5).

Statistical analyses also revealed a context main effect. There was a greater ERP amplitude in the same-context condition (average of the ERPs of all the items presented within the same context) compared to the different-context condition (average of the ERPs of all the items presented within a different context) between 300 and 500 ms (*p* < 0.05; d’ = 0.48) and also between 500 and 800 ms (*p* < 0.05; d’ = 0.51). In the early time window, we found a cluster of centroposterior electrodes (*p* < 0.05), especially on the left side (Figure 6: left). In the late temporal window (500–800 ms), we observed a cluster of central-anterior electrodes (*p* < 0.05). (Figure 6: right). The category by context interaction did not reach statistical significance (*p* > 0.05).

No statistical differences were found in the first item analysis in any of the relevant temporal windows. The last item analysis revealed no significant differences between the five response conditions, but event-related potential amplitudes were significantly higher in the same-context level in early (*p* < 0.05; d’ = 0.47) and late (*p* < 0.05; d’ = 0.45) temporal windows. The first temporal window (300–500 ms) revealed a cluster of centroposterior electrodes (*p* < 0.05) (Figure 7: left). Similarly, the late temporal window (500–800 ms) revealed a cluster of centroposterior electrodes (*p* < 0.05) (Figure 7: right).

## 4. Discussion

In the current study, we sought to determine the effects of context stability at encoding on mnemonic discrimination and its neural substrate [38,39]. To the best of our knowledge, this is the first study focusing on the encoding phase. Previous studies focused on the recognition stage and showed that keeping the background context consistent at the encoding and recognition phases increased correct target recognition, but also led to false recognition of lure items (i.e., similar items that were never actually learned), thus reducing the ability to discriminate (i.e., behavioral pattern separation) [19,20,22,23,24,25]. This modulation of background context on recognition and discrimination has been interpreted as an increase in familiarity due to the reappearance of encoding context, which aids target identification, but at the same time reduces mnemonic discrimination due to the increase in familiarity of the similar lure items [21,24]. This interpretation appears to be supported by the results from Libby, Reagh, Bouffard, Ragland, and Ranganath [22], who showed that hippocampal activity patterns were different for similar elements that had different encoding contexts but overlapped for similar elements that shared contextual information. Likewise, Herz, Bukala, Kragel, and Kahana [20] found that false recalls that shared greater contextual similarity with the target context were associated with a hippocampal low-frequency activity reduction, similar to the reduction associated with correct recalls. Contrary to these studies, we did not find a significant modulation of background context on target identification, or mnemonic discrimination performance. Our results aligned with those of Bouffard, Fidalgo, Brunec, Lee, and Barense [54], who tested how the distinctiveness of objects or scenes aided memory. Participants studied 34 scene–object pairs under three conditions: distinct scenes paired with similar objects, similar scenes paired with distinct objects, and similar scenes paired with similar objects. After the study phase, participants performed a single-item recognition test (a single image, either a scene or an object), followed by an associative memory judgment. They found that, regardless of whether objects and scenes were similar or distinct, participants showed intact single-item recognition of scenes and objects, which suggests that they rely on distinct objects (not scenes) to distinguish between similar memories (see also [55]). Likewise, other authors [56] provided evidence suggesting that reinstatement of content- and context-based information occurs within separate cortical circuits so that semantic representations can cue memories in a context-independent manner. Similarly, Stevenson, Reagh, Chun, Murray, and Yassa [36] proposed that source memory and pattern separation are separable processes that might be supported by distinct neural mechanisms. Gronau and Shachar [38] also reported that when using relatively long exposure durations during encoding (i.e., 2 s), the influence of contextual information on recognition is eliminated. Additionally, Palmer, Grilli, Lawrence, and Ryan [23] showed that participants were worse at identifying similar objects when they were placed in a scene context, repeated or novel, compared to a repeated white background. Finally, it should also be taken into account that context influence may change or even reverse depending on the presence or absence of contextual cues during recall [39]. Although we did not find a significant influence of background context, we observed a main effect of category. Remarkably, mnemonic discrimination took place, since similar lure correct rejections were comparable to correct recalls and higher than the rest of the response types (Figure 2). Additionally, old hits were statistically different from similar FAs (marking a lure as old), suggesting that participants were able to effectively differentiate between old items and lures. Altogether, these results support the idea of successful target recognition and behavioral pattern separation between similar items [33,57]. Previous studies have shown that participants are commonly able to correctly identify targets as ‘old’ and foils as ‘new’ in a high proportion of cases, while having more difficulties in identifying lures as ‘similar’ [17,57]. Interestingly, it seems that participants in our study were more likely to classify old items as similar than previously reported. Further research is needed to fully understand this discrepancy.

In addition, reaction time analyses showed that similar CRs were significantly slower than old false alarms, old hits, similar FAs, and new CRs. Moreover, old false alarms were slower than old hits and new CRs. These results are consistent with the previous literature, such as the works of García-Rueda, Poch, and Campo [10], who found significantly slower reaction times for similar CRs in comparison with old hits, similar FAs, and new CRs. This slower reaction time for similar CRs suggests that discriminating between items within a category is a more complex process than recognizing previously encoded items or items from an unseen category. Hayes, Nadel, and Ryan [29] similarly found that participants were slowest to respond to the “scene lures” (similar to targets, but novel object presented in a novel scene) than any other condition, supporting the idea that distinguishing related items and scenes involves a more complex processing effort. Moreover, pairwise comparisons between memory judgment (correct or incorrect) and conditions revealed that participants were faster to respond to “object.object” correct trials (old object presented on a white background during encoding and testing phases), followed by “scene.scene” correct trials (old object presented on an old background). Participants were slowest to respond to the “scene.object” correct trials (an old object without a background that was previously presented with a background during the study), which did not differ in response times from object lure correct rejections (similar to targets, but novel object on a white background) and scene lure correct rejections. These results are also consistent with our study, as they suggest that participants experienced greater difficulty in discriminating the object when the context was lost and that this difficulty was comparable to correctly rejecting a lure.

One limitation of the study relates to the absence of on-screen reminders of the response options during the trials. Participants were verbally instructed on the key–response associations, and these instructions were displayed on the screen before the test, as well as during breaks. In contrast, these instructions were not displayed during the trials to minimize distractions. However, this design may have affected performance due to the potential reliance on memory for key–response associations during trials. This potential issue could be addressed in future studies by presenting on-screen reminders during the trials.

### ERP Findings

Participants processed images belonging to different categories during the study phase (four per category). ERPs from the encoding of image categories (when averaging all items) (see [31]) that were correctly identified as old (old hits) were equivalent to the ERPs for those categories that were subsequently marked as similar (old false alarms indicates recognition of the category) and for those categories that were lures which attracted old responses (similar false alarms). Furthermore, ERPs from the encoding of image categories that were subsequently recognized (old hit) were significantly different from those categories that were subsequently marked as new (new miss indicates both category and item non-recognition). These results suggest a category recollection in which the consecutive presentation of similar items within a category created a strong category-related memory trace [20,31]. This trace is reflected in an increased ERP positivity of those categories that were previously seen and recognized in comparison to non-recognized and marked-as-new items (new miss). Therefore, the reduction in ERP positivity for new misses may be indicative of less effective memory encoding strength and consequently a weaker category-related memory trace. These results are consistent with global matching models, which propose that the memory strength of a tested item arises from the similarity between its representation and all other representations from studied items (known as global similarity) [13]. Thus, higher neural global similarity during encoding leads to an increase in recognition memory [7,8,13]. However, the strengthening of categories could also lead to an increased ERP positivity of similar false alarms, as the items were new but belonged to previously seen categories. In this way, old hits and similar FAs had a more positive ERP than new misses (Figure 5), although only the old hits showed differences at a statistical level. ERPs from encoding category images (when averaging all items) from the same-context group had higher activation than categories from the different-context group in both early and late temporal windows, respectively, associated with familiarity and recollection. These results suggest a stronger memory trace for those category images that have the same inter-category context, regardless of the response (old, similar, or new). Thus, common context during encoding could facilitate the formation of a stronger memory for images within a category and support the idea that associative memory formation is facilitated by similarity across encoding patterns [58]. However, this ERP increase could only be reflecting the recognition of the context, without implying a strengthening of the item associated with it, so in the absence of context, this recognition of common contexts would not necessarily translate into an advantage in remembering specific items that were in common contexts. Additionally, previous research has shown that recollection can occur in the absence of pattern separation. Specifically, while accurate recollection might require pattern separation, false recollection might not [59]. In another experiment, Hayes, Nadel, and Ryan [29] found that recognition of context-free objects that were previously studied with a scene was lower than recognition of objects that were previously studied with a white background or context-presented objects that were previously studied with that scene. Therefore, it could be concluded that context is an element of the single episodic trace that is not determinant to recover a specific item, although it could be used as a cue when presented and even decrease the item recognition when it is changed, compared to those conditions that continue with the same backgrounds (both white and visually rich scenes) [38]. Consistently, Dohm-Hansen and Johansson [19] designed an experiment where they presented pictures of objects in different contexts. During the test, they presented objects and contexts in different conditions (old object–old context, new object–new context, similar object–similar context, old object–similar context, and similar object–old context) and participants had to respond whether the object and context were old, similar, or new. The results showed that the hit rate for objects was higher when the accompanying context had been presented previously than when the context was a lure. Contextual information may therefore influence object recall. Similarly, participants had a higher hit rate for contexts when the accompanying objects had been previously presented compared to when they were lures, so the object may also influence the ability to remember the context. ERPs of the encoding of the first presented items (the item that is subsequently tested in the discrimination phase) that were subsequently recognized did not differ from either the old false alarm or the new miss in any of the contexts. Taken together, the results of the category condition and the first item condition suggest that it is not the encoding of the tested item, but the encoding of all the studied items, that influences subsequent recognition through familiarity with the whole object category [31,60]. ERPs of the encoding of the fourth presentation (the last item studied within each category) showed both an increase in the amplitude for the common contexts and no differences between the category responses, which was added in a consistent manner with the non-differentiation between old hits and similar false alarms, indicating that there was a recognition of the category. The previous literature suggests that the hippocampus enhances differences between events, even though they share item or context information ([9,13], although see [22,61]). Instead, there is also previous research suggesting that, although neural coding in the hippocampus may differentiate between events with some overlapping attributes, it assigns overlapping neural representations when the average amount of overlap between stimuli is high [62].

Overall, our results suggest a category-based recognition, as old hits did not differ from old false alarms or similar FAs during encoding. One possible explanation is that episodic memory formation is a single process, meaning that different forms of memory, which contribute to different aspects of the trace (such as context), reflect differing levels of a single encoding mechanism that forge distinct object and contextual representations into a coherent episodic trace [4]. This way, the increased amplitude of the ERPs from the same-context condition could reflect a repetition factor that is not determinant to recover the item trace at the behavioral level. Additionally, we should mention that other factors such as attentional deployment, working memory capacity, cognitive flexibility, and decisional processes could be influencing current results [63,64,65,66], so further research is warranted. Furthermore, as has been recently stated, pattern separation is best considered the result of network interactions [15,67], so exploring cortico-hippocampal networks could provide further understanding.

## 5. Conclusions

Our results suggest that behavioral pattern separation can take place even when there is a similarity of context and item between events. The cortical activity also suggests that the encoding of all items studied, rather than the encoding of the subsequently tested item, influences subsequent recall. Finally, the differences in cortical activity between contexts at encoding suggest a process of familiarity through repetition of the same context that is not determinant in behavioral discrimination. These differences thus support the idea that different forms of memory are recruited during encoding to build a single episodic trace.

## Figures and Tables

**Figure 1 neurosci-06-00013-f001:**
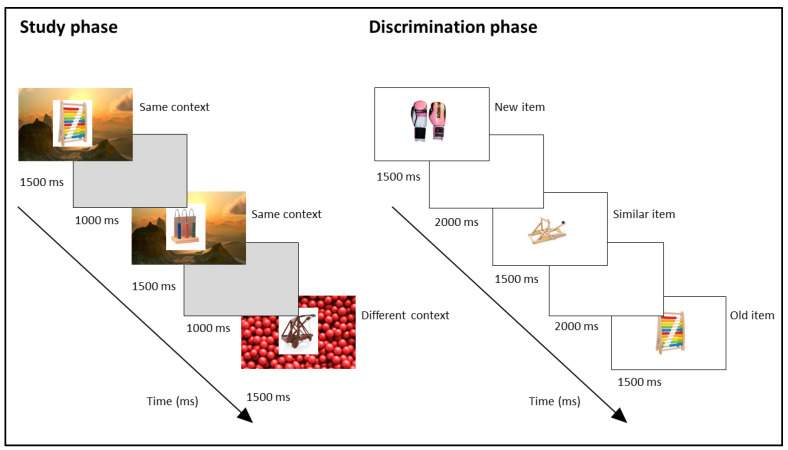
Test structure and examples of the items employed. The 1280 images of objects from different categories were presented at the study phase. Each image was presented for 1500 ms, followed by a 1000 ms gray screen. The similarity of the context presented for each category was manipulated to “same” or “different” context. Twenty minutes after the study phase, the discrimination phase took place. During the task, participants were shown 400 images of objects and were instructed to classify each image as old (previously seen in the study phase), similar (new object, but from a category previously seen), or new (new object from a new category). Each image appeared for 1500 ms and was followed by a 2000 ms white screen.

**Figure 2 neurosci-06-00013-f002:**
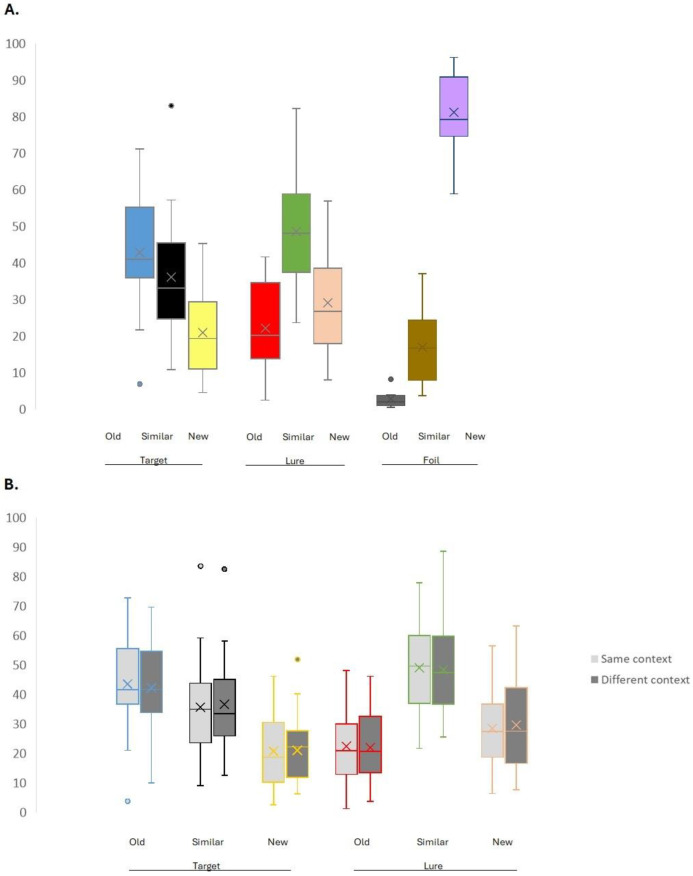
(**A**) Mean proportion of old, similar, and new responses to targets, lures and foils. (**B**) Mean proportion of old, similar, and new responses to targets from same or different contexts and lures from same or different contexts. The cross inside the box indicates the mean.

**Figure 3 neurosci-06-00013-f003:**
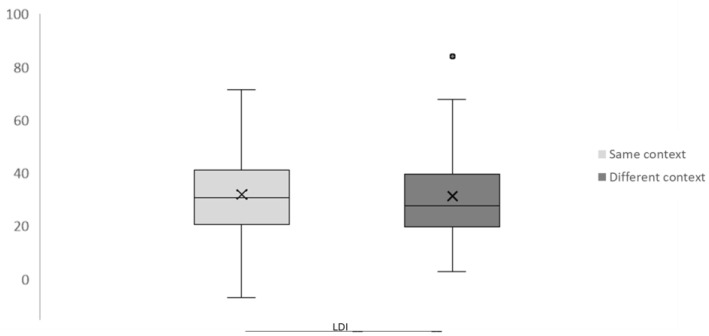
Lure discrimination index. Proportion of correctly identified lures corrected for the baseline rate of similar responses to novel items in terms of context. The cross inside the box indicates the mean.

**Figure 4 neurosci-06-00013-f004:**
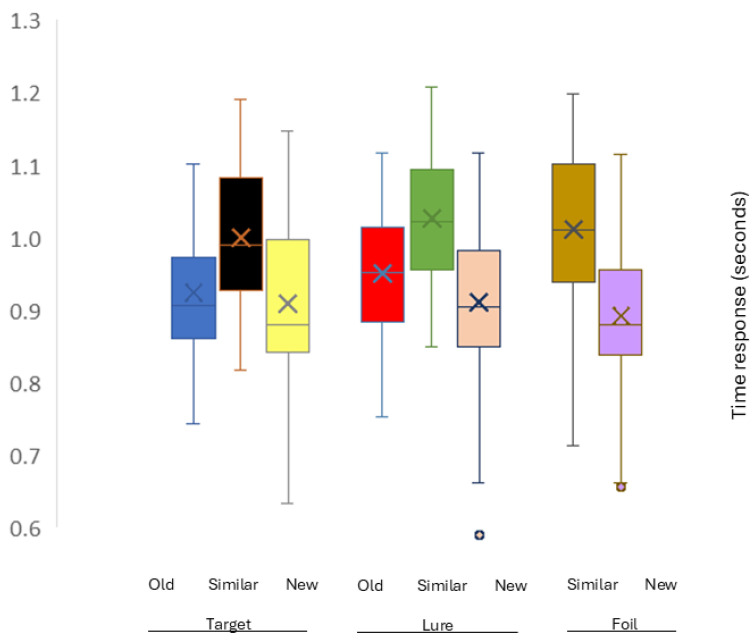
Reaction time responses of old, similar, and new responses to targets, lures, and foils.

**Figure 5 neurosci-06-00013-f005:**
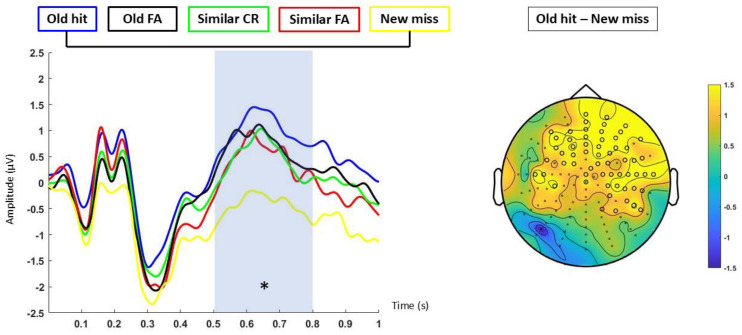
Encoding, category analysis (response). Differences reaching statistical significance are marked with an asterisk in the ERPs and with black circles in the topography. Left: ERPs from “old hit” (blue line), “similar FA” (red line), “similar CR” (green line), “old FA”, and “new miss” in the 500–800 ms time window. Right: topographic map of the grand average of “old hits–new misses” in the 500–800 ms time window.

**Figure 6 neurosci-06-00013-f006:**
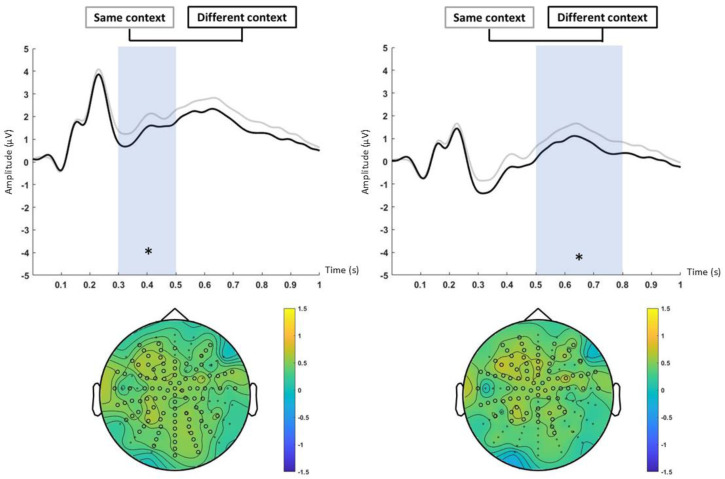
Encoding, category analysis (context). Significant differences are marked with an asterisk in the ERPs and with black circles in the topography. Top and left: ERPs from “same context” (gray line), and “different context” (black line) in the 300–500 ms time window. Bottom and left: topographic map of the grand average of “same context–different context” in the 300–500 ms time window. Top and right: ERPs from “same context” (gray line), and “different context” (black line) in the 500–800 ms time window. Bottom and right: topographic map of the grand average of “same context–different context” in the 500–800 ms time window.

**Figure 7 neurosci-06-00013-f007:**
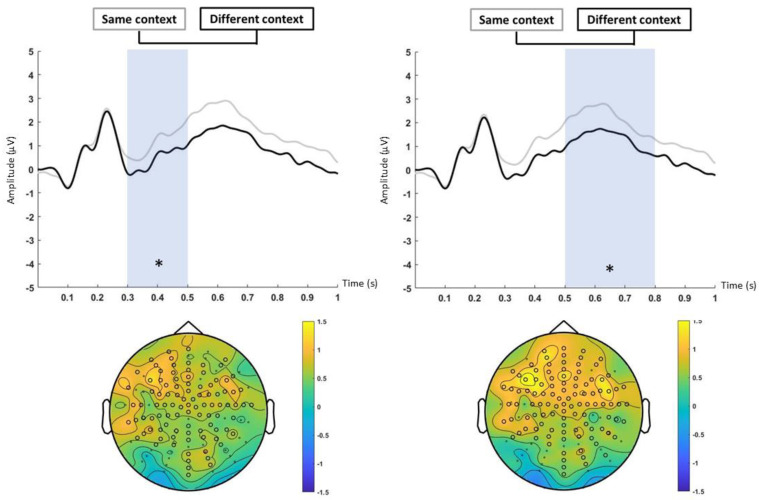
Encoding, last item analysis (context). Significant differences are marked with an asterisk in the ERPs and with black circles in the topography. Top and left: ERPs from “same context” (gray line), and “different context” (black line) in the 300–500 ms time window. Bottom and left: topographic map of the grand average of “same context–different context” in the 300–500 ms time window. Top and right: ERPs from “same context” (gray line), and “different context” (black line) in the 500–800 ms time window. Bottom and right: topographic map of the grand average of “same context–different context” in the 500–800 ms time window.

**Table 1 neurosci-06-00013-t001:** Mean proportion of Responses and Significant Differences.

Table Mean Proportion of Responses	M	SD	Significant Differences *
Old response to target (1)	42.900	2.947	2, 3, 6, 7, 9
Old response to lure (2)	22.209	2.245	1, 3, 5, 9
Old response to foil (3)	1.887	0.473	1, 2, 4, 5, 6, 7, 8, 9
Similar response to target (4)	36.167	3.093	3, 5, 6, 9
Similar response to lure (5)	48.674	3.004	2, 3, 4, 6, 7, 8, 9
Similar response to foil (6)	16.930	1.838	1, 3, 4, 5, 9
New response to target (7)	20.933	2.258	1, 3, 5, 8, 9
New response to lure (8)	29.117	2.692	3, 5, 7, 9
New response to foil (9)	81.183	2.035	1, 2, 3, 4, 5, 6, 7, 8

* See Appendix A for pairwise comparisons.

**Table 2 neurosci-06-00013-t002:** Reaction time responses and Significant Differences.

Reaction Time Responses	M	SD	Significant Differences *
Old response to target (1)	0.962	0.083	4, 5
Old response to lure (2)	0.986	0.084	4
Similar response to target (3)	1.030	0.087	6, 4, 7, 8
Similar response to lure (4)	1.054	0.079	1, 3, 6, 7, 8
Similar response to foil (5)	1.040	0.099	1, 6, 7, 8
New response to target (6)	0.955	0.114	3, 4, 5
New response to lure (7)	0.955	0.114	3, 4, 5
New response to foil (8)	0.933	0.100	1, 6, 7, 8

* See Appendix B for pairwise comparisons.

## Data Availability

Data will be made available on request.

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
