# Peer review of "Contextual Influence on Pattern Separation During Encoding"

_neurosci, 2025, doi:10.3390/neurosci6010013_

Round 1

Reviewer 1 Report (Previous Reviewer 2)

Comments and Suggestions for Authors

I thank the authors for responding to our requests. I believe the manuscript has improved significantly. However, there are minor issues that could be considered:

Regarding the major concerns:

2.2 The box-and-whisker plots give a better idea of the distribution of the individual data. The only suggestion about these plots is to indicate in the captions the meaning of the cross inside the box. I assume it represents the mean, but it should be explicitly indicated.

Regarding the minor concerns:

3.1 The explanation about the "New hit" EEG register was clear and understandable. However, the issue about category-related memory trace and its relation to a smaller amplitude ERP is still unanswered. It would be good if the authors could add a consideration about this in the discussion section.

2 & 3. As the authors proposed, it would be very valuable to add the information to the methodology section for additional clarity and, in the case of point 2, discuss the limitation introduced to the study by the described situation.

Author Response

Reviewer 1

I thank the authors for responding to our requests. I believe the manuscript has improved significantly. However, there are minor issues that could be considered:

Regarding the major concerns:

2.2 The box-and-whisker plots give a better idea of the distribution of the individual data. The only suggestion about these plots is to indicate in the captions the meaning of the cross inside the box. I assume it represents the mean, but it should be explicitly indicated.

We appreciate the valuable suggestion regarding the box-and-whiskers plots. We updated the captions to explicitly indicate the meaning of the cross inside the box. As correctly assumed, the cross represents the mean of the data.

“Figure 2. A. Mean proportion of old, similar and new responses to targets, lures and foils. B. Mean proportion of old, similar and new responses to targets from same or different context and lures from same or different context. The cross inside the box indicates the mean.

“Figure 3. Lure Discrimination Index. Proportion of correctly identified lures corrected for the baseline rate of similar responses to novel items in terms of context. The cross inside the box indicates the mean.”

Regarding the minor concerns:

3.1 The explanation about the "New hit" EEG register was clear and understandable. However, the issue about category-related memory trace and its relation to a smaller amplitude ERP is still unanswered. It would be good if the authors could add a consideration about this in the discussion section.

We appreciate the comment and apologize for the lack of clarity. We updated the discussion section to explicitly address the connection between category-related memory traces and smaller amplitude ERPs for new misses.

“Furthermore, ERPs from the encoding of image categories that were subsequently recognized (old hit) were significantly different from those categories that were subsequently marked as new (new miss indicates both category and item non-recognition). These results suggest a category recollection in which the consecutive presentation of similar items within a category created a strong category-related memory trace [20,31]. This trace is reflected in an increased ERP positivity of those categories which were previously seen and recognized in comparison to non-recognized and marked-as-new items. Therefore, the reduction in ERP positivity for new misses may be indicative of less effective memory encoding strengthening and consequently a weaker category-related memory trace. These results are consistent with global matching models, which propose that the memory strength of a tested item arises from the similarity between its representation and all other representations from studied items (known as global similarity) [13]. Thus, higher neural global similarity during encoding leads to an increase in recognition memory [7-8,13]. However, strengthening of categories could also lead to an increased ERP positivity of similar false alarms, as the items were new but belonged to previously seen categories. In this way, old hit and similar FA had a more positive ERP than the new miss (Figure 5), although only the old hits showed differences at a statistical level.”

2 & 3. As the authors proposed, it would be very valuable to add the information to the methodology section for additional clarity and, in the case of point 2, discuss the limitation introduced to the study by the described situation.

We appreciate the suggestion and, as requested, we added the relevant information to the methodology section for greater clarity. In the case of point 2, we have also discussed the limitations introduced by the absence of on-screen reminders in the discussion section of the manuscript.

“Participants were verbally instructed to press key numbers 1, 2, or 3 while the image is on the screen to indicate whether the items were old, lure, or new, respectively. Additionally, these instructions were displayed on the screen before starting the test. During the trials, these response options were not displayed on the screen, but during each break, the instructions were shown again on the screen to remind participants of the correct key-response associations, so they did not have to rely on memory during the break. Participants were given a non-timed pause at the midpoint of the discrimination phase, after the first 200 images, to reduce fatigue and maintain focus. The break was untimed to allow participants to rest adequately and resume when they felt ready, ensuring that performance in the second half of the task was not compromised by fatigue or time pressure. The test phase lasted approximately 24 minutes, so it was divided into two blocks of 12 minutes each. During the encoding phase, the same criterion was followed by dividing the study into blocks of 320 pictures, so each block lasted approximately 13 minutes. We took into account previous studies that suggest the performance decline typically occurs within the first 15 minutes [46-52]”.

One limitation of the study relates to the absence of on-screen reminders of the response options during the trials. Participants were verbally instructed on the key-response associations, and these instructions were displayed on the screen before the test, as well as during breaks. In contrast, these instructions were not displayed during the trials to minimize distractions. However, this design may have affected performance due to potential reliance on memory for key-response associations during trials. This potential issue could be addressed in future studies by presenting on-screen reminders during the trials.”

Reviewer 2 Report (New Reviewer)

Comments and Suggestions for Authors

The manuscript entitled "Contextual Influence on Pattern Separation During Encoding" presents a well-structured investigation into the effects of "context stability" during encoding on mnemonic discrimination and its neural correlates. The study contributes valuable insights into how ERPs may constitute valuable biomarkers in understanding temporal dynamics during memory encoding. While the study design is robust and the results rather novel and interesting, there are several points that should be revised in order to improve the clarity of the study and make it acceptable for publication.

SPECIFIC COMMENTS

(1) The term "context stability" should be unequivocally be defined. In particular is this term equivalent to "consistency of background context". Please specify and use the terms in order to avoid potential misleading definitions.

(2) Attention and executive functions are integral to encoding and pattern separation processes. The Authors should clarify whether variations in attentional engagement (e.g., reaction times, eye-tracking data, or attention-related ERP components like P3) were controlled or analyzed during the task.

(3) The ERP results do not report any quantitative analysis of the latencies and amplitudes of the main wave components for the various response conditions. This analysis is absolutely necessary. For instance, the authors should consider the book [R1], which provides foundational insights into the theory, methodology, and applications of ERP techniques, including components like P3 and their relationship to attention. Additionally [R2] offers a detailed analysis of P3 variations in response to attentional demands, which might be relevant here.

(4) The lack of significant context effects at the behavioral level warrants deeper analysis, particularly in relation to executive functions such as working memory and cognitive flexibility, which may influence ERP outcomes. Findings on the transformation of experience into memory [R3], as well as on hippocampal-cortical functional connectivity during encoding [R4], could offer theoretical frameworks to better interpret these results.

(5) About the integration of behavioral and neural findings, the authors might discuss how do the ERP differences between contexts explain the observed behavioral pattern separation?  The relationship between cognitive load and attention modulation has been emphasized by ERP dynamics during a dual n-back task in ADHD and control populations [R5]. Those findings could offer a comparative perspective to contextualize the current study's encoding-related ERP effects.

(6) The authors should critically evaluate how their findings align or contrast with previous ERP research. In the Discussion, the authors could provide a richer context by contrasting their results with the work of Amer and Davachi (2023) or Libby et al. (2019). Additionally, the ERP dynamics reported in the current study, particularly regarding the role of cortical networks during encoding, could be better contextualized by citing an analysis of brain networks underlying episodic memory retrieval [R6].

(7) The interpretation of ERP findings as reflecting "familiarity" and "recollection" processes is plausible but not definitively supported. The authors should acknowledge alternative explanations and discuss the potential role of extra-hippocampal regions in driving these effects.

(8) Figures 5 and 6 are informative but require improved labels and legends. For instance, explicitly marking the conditions compared and highlighting the significant clusters would aid readability.

(9) Supplementary figures showing individual ERP waveforms for key electrodes would provide additional transparency.

(10) The statistical analysis is incomplete. Besides all p-values, the authors should report the corresponding effect sizes.

(11) The authors should conduct exploratory analyses to identify potential moderating variables (e.g., individual differences in working memory capacity) that could explain the null context effect. Moreover, the Discussion should expand on the implications of the null context effect and its alignment with existing theories of episodic memory.

(12) Future studies might consider how combining ERP and fMRI could provide a more comprehensive understanding of pattern separation during encoding.

Cited References.

[R1] Luck SJ. (2005). An Introduction to the Event-Related Potential Technique. MIT Press.

[R2] Kok A. (2001) On the utility of P3 amplitude as a measure of processing capacity. Psychophysiology 38(3):557-77. doi: 10.1017/s0048577201990559

[R3] Paller KA, Wagner AD. (2002) Observing the transformation of experience into memory. Trends Cogn Sci. 6(2):93-102. doi: 10.1016/s1364-6613(00)01845-3

[R4] Raud L, Sneve MH, Vidal-Piñeiro D, Sørensen Ø, Folvik L, Ness HT, Mowinckel AM, Grydeland H, Walhovd KB, Fjell AM. (2023) Hippocampal-cortical functional connectivity during memory encoding and retrieval. Neuroimage 279:120309. doi: 10.1016/j.neuroimage.2023.120309

[R5] Lintas A, Mesrobian SK, Bader M, Villa AEP. (2021) ERPs in controls and ADHD patients during dual n-back task. Advances in cognitive neurodynamics (VII), pp. 189–203. Springer Nature Singapore. doi: 10.1007/978-981-16-0317-4_20

[R6] Rugg MD, Vilberg KL. (2013) Brain networks underlying episodic memory retrieval. Curr Opin Neurobiol. 23(2):255-60. doi: 10.1016/j.conb.2012.11.005

Author Response

Reviewer 2

The manuscript entitled "Contextual Influence on Pattern Separation During Encoding" presents a well-structured investigation into the effects of "context stability" during encoding on mnemonic discrimination and its neural correlates. The study contributes valuable insights into how ERPs may constitute valuable biomarkers in understanding temporal dynamics during memory encoding. While the study design is robust and the results rather novel and interesting, there are several points that should be revised in order to improve the clarity of the study and make it acceptable for publication.

SPECIFIC COMMENTS

(1) The term "context stability" should be unequivocally be defined. In particular is this term equivalent to "consistency of background context". Please specify and use the terms in order to avoid potential misleading definitions.

We thank the reviewer for this suggestion. According to the suggestion we have substituted the term "context stability" by the clearer term "consistency of background context".

(2) Attention and executive functions are integral to encoding and pattern separation processes. The Authors should clarify whether variations in attentional engagement (e.g., reaction times, eye-tracking data, or attention-related ERP components like P3) were controlled or analyzed during the task.

We agree with the reviewer about the crucial role of attention and executive functions in memory processes. We did not register eye movements, beyond the control of blinks and horizontal movements, nor analyzed P3 wave. We recorded participants’ reaction times, as can be found in the Results and Discussion sections.

(3) The ERP results do not report any quantitative analysis of the latencies and amplitudes of the main wave components for the various response conditions. This analysis is absolutely necessary. For instance, the authors should consider the book [R1], which provides foundational insights into the theory, methodology, and applications of ERP techniques, including components like P3 and their relationship to attention. Additionally [R2] offers a detailed analysis of P3 variations in response to attentional demands, which might be relevant here.

The ERP results report quantitative analyses in our two-time windows of interest, chosen based on our previous studies (Liu et al., 2017; Bai et al., 2015; Kamp & Zimmer, 2015; Kamp, Bader, & Mecklinger, 2017). One of the strengths of the statistical analysis used here (cluster‐based permutation tests) is that it reports the significant differences between the compared conditions without preselecting specific electrodes. Cluster‐based permutation analyses have gained an almost universal acceptance as inferential procedures in cognitive neuroscience, and we consider they are the better analytical tool for the hypotheses we aim to test, based on our previous work—please note that even the author of [R1], Steven Luck, has subscribed to the claim that ‘… non-parametric permutation-based approaches … currently outperform other methods [in dealing with multiple comparisons]’ (Niso et al., 2022, in NeuroImage, with Luck as co-author). Although the analysis proposed by the reviewer could seem complementary, we feel it falls outside of the scope of our study and provides suboptimal comparability with previous research, as opposed to our method of choice.

  • Yi Liu, Timm Rosburg, Chuanji Gao, Christine Weber, Chunyan Guo. Differentiation of subsequent memory effects between retrieval practice and elaborative study. Biological Psychology, Volume 127, 2017, Pages 134-147, ISSN 0301-0511. https://doi.org/10.1016/j.biopsycho.2017.05.010.
  • Bai C-H, Bridger EK, Zimmer HD and Mecklinger A (2015) The beneficial effect of testing: an event-related potential study. Front. Behav. Neurosci. 9:248. doi: 10.3389/fnbeh.2015.00248
  • Kamp S.-M., Zimmer H. D. (2015). Contributions of attention and elaboration to associative encoding in young and older adults. Neuropsychologia 75 252–264. 10.1016/j.neuropsychologia.2015.06.026
  • Kamp, S.-M., Bader, R., & Mecklinger, A. (2017). ERP Subsequent Memory Effects Differ between Inter-Item and Unitization Encoding Tasks. Frontiers in Human Neuroscience, 11. https://doi.org/10.3389/fnhum.2017.00030

(4) The lack of significant context effects at the behavioral level warrants deeper analysis, particularly in relation to executive functions such as working memory and cognitive flexibility, which may influence ERP outcomes. Findings on the transformation of experience into memory [R3], as well as on hippocampal-cortical functional connectivity during encoding [R4], could offer theoretical frameworks to better interpret these results.

We thank the reviewer for this comment. However, while we did not consider WM nor cognitive flexibility in our experimental design, unfortunately we do not have any information regarding these functions. Nonetheless, we have rewritten the Discussion and included a new paragraph highlighting these issues.

“Additionally, we should mention that other factors such as attentional deployment, working memory capacity, cognitive flexibility and decisional processes could be influencing current results (deBettencourt et al., 2017; Long, 2023; Paller & Wagner, 2002; Schneider, s et al., 2024), so further research is warranted. Furthermore, as has been recently stated, pattern separation is best considered the result of network interactions (Amer & Davachi, 2023; Rugg MD, Vilberg, 2013), so exploring cortico-hippocampal networks could provide further understanding.”

(5) About the integration of behavioral and neural findings, the authors might discuss how do the ERP differences between contexts explain the observed behavioral pattern separation?  The relationship between cognitive load and attention modulation has been emphasized by ERP dynamics during a dual n-back task in ADHD and control populations [R5]. Those findings could offer a comparative perspective to contextualize the current study's encoding-related ERP effects.

The reviewer refers to cognitive load in a WM paradigm, which is not the case at hand. We have already provided an interpretation of the results, but it is not possible to relate it to WM load since we did not consider this dimension in our experimental paradigm, and then it could turn into an elucubration.

(6) The authors should critically evaluate how their findings align or contrast with previous ERP research. In the Discussion, the authors could provide a richer context by contrasting their results with the work of Amer and Davachi (2023) or Libby et al. (2019). Additionally, the ERP dynamics reported in the current study, particularly regarding the role of cortical networks during encoding, could be better contextualized by citing an analysis of brain networks underlying episodic memory retrieval [R6].

We thank the reviewer for this comment. We have rewritten the discussion accordingly.

“Furthermore, as has been recently stated, pattern separation is best considered the result of network interactions (Amer & Davachi, 2023; Rugg MD, Vilberg, 2013), so exploring cortico-hippocampal networks could provide further understanding.”

(7) The interpretation of ERP findings as reflecting "familiarity" and "recollection" processes is plausible but not definitively supported. The authors should acknowledge alternative explanations and discuss the potential role of extra-hippocampal regions in driving these effects.

We thank the reviewer for raising this point. We have rewritten the discussion to include alternative explanations as suggested.

“Additionally, previous research has shown that recollection can occur in the absence of pattern separation. Specifically, while accurate recollection might require pattern separation, false recollection might not (Kim and Yassa, 2013).” 

(8) Figures 5 and 6 are informative but require improved labels and legends. For instance, explicitly marking the conditions compared and highlighting the significant clusters would aid readability.

We thank the reviewer for the suggestion and have added additional labels and legends to Figures 5, 6 and 7 to improve their clarity. We hope these changes have enhanced the quality of the manuscript.

Figure 5. Encoding, Category analysis (Response). Differences reaching statistical significance are marked with an asterisk in the ERPs and with black circles in the topography. Left: ERPs from “old hit” (blue line), “similar FA” (red line), “similar CR” (green line), “old FA”, and “new miss” in the 500-800 ms time window. Right: topographic map of the grand-average of “old hits-new miss” in the 500-800 ms time window.

Figure 6. Encoding, Category analysis (Context). Significant differences are marked with an asterisk in the ERPs and with black circles in the topography. Top and Left: ERPs from “same context” (grey line), and “different context” (black line) in the 300-500 ms time window. Bottom and Left: topographic map of the grand-average of “same context-different context” in the 300-500 ms time window. Top and Right: ERPs from “same context” (grey line), and “different context” (black line) in the 500-800 ms time window. Bottom and Right: topographic map of the grand-average of “same context-different context” in the 500-800 ms time window.

Figure 7. Encoding, Last Item analysis (Context). Significant differences are marked with an asterisk in the ERPs and with black circles in the topography. Top and Left: ERPs from “same context” (grey line), and “different context” (black line) in the 300-500 ms time window. Bottom and Left: topographic map of the grand-average of “same context-different context” in the 300-500 ms time window. Top and Right: ERPs from “same context” (grey line), and “different context” (black line) in the 500-800 ms time window. Bottom and Right: topographic map of the grand-average of “same context-different context” in the 500-800 ms time window.

(9) Supplementary figures showing individual ERP waveforms for key electrodes would provide additional transparency.

We appreciate the reviewer’s suggestion and prepared supplementary figures that show individual ERP waveforms for key electrodes. These figures display the ERPs for each participant in every experimental condition. As the reviewer would note, there is one participant whose data deviates from the rest. However, we repeated the analyses excluding this participant, and the results remained unchanged. We also offer the possibility of sharing the data to further increase transparency. Thank you for your valuable comments.

Below are the individual ERPs for:

Old hitSimilar FA

Similar CR

Old FA

New miss

Same context (300-500 ms time window)

Different context (300-500 ms time window)

Same context (500-800 ms time window)

Different context (500-800 ms time window)

(10) The statistical analysis is incomplete. Besides all p-values, the authors should report the corresponding effect sizes.

We appreciate the reviewer’s suggestion and have added the corresponding effect sizes in the statistical analysis, along with the p-values, as requested.

“The Category analysis showed that ERP amplitudes were significantly different between 500 and 800 ms (p < .05; d’ = .29). This effect was explained by a more positive deflection in a central-anterior cluster of electrodes for the old hits compared to the new misses responses, which was more evident on the right side (p < .05). (Figure 5).

Statistical analyses also revealed a context main effect. There was a greater ERP amplitude in the same context condition (average of the ERPs of all the items presented within the same context) compared to the different context condition (average of the ERPs of all the items presented within the different context) between 300 and 500 ms (p < .05; d’ = .48) and also between 500 and 800 ms (p < .05; d’ = .51). In the early time window, we found a cluster of centroposterior electrodes (p < .05), specially on the left side (Figure 6: Left). In the late temporal window (500-800 ms), we observed a cluster of central-anterior electrodes (p <.05). (Figure 6: Right). The Category by Context interaction did not reach statistical significance (p > .05).

No statistical differences were found in the First Item analysis in any of the relevant temporal windows. The Last Item analysis revealed no significant differences between the five response conditions, but event-related potential amplitudes were significantly higher in the same context level in early (p < .05; d’ = .47) and late (p < .05; d’ = .45) temporal windows. The first temporal window (300-500 ms) revealed a cluster of centroposterior electrodes (p < .05) (Figure 6: Left). Similarly, the late temporal window (500-800 ms) revealed a cluster of centroposterior electrodes (p < .05) (Figure 6: Right).”

(11) The authors should conduct exploratory analyses to identify potential moderating variables (e.g., individual differences in working memory capacity) that could explain the null context effect. Moreover, the Discussion should expand on the implications of the null context effect and its alignment with existing theories of episodic memory.

While we agree with the reviewer, since we have not used measures of WM capacity it is impossible to conduct the suggested analyses. We will consider them in future studies and thank the reviewer for pointing out these issues.

(12) Future studies might consider how combining ERP and fMRI could provide a more comprehensive understanding of pattern separation during encoding.

We totally agree with the reviewer and thank for the insightful suggestion. An interesting avenue for future research could involve combining ERP and fMRI methodologies to further explore pattern separation during encoding. While ERPs provide excellent temporal resolution for identifying neural responses, fMRI could complement this by offering precise spatial resolution.

Cited References.

[R1] Luck SJ. (2005). An Introduction to the Event-Related Potential Technique. MIT Press.

[R2] Kok A. (2001) On the utility of P3 amplitude as a measure of processing capacity. Psychophysiology 38(3):557-77. doi: 10.1017/s0048577201990559

[R3] Paller KA, Wagner AD. (2002) Observing the transformation of experience into memory. Trends Cogn Sci. 6(2):93-102. doi: 10.1016/s1364-6613(00)01845-3

[R4] Raud L, Sneve MH, Vidal-Piñeiro D, Sørensen Ø, Folvik L, Ness HT, Mowinckel AM, Grydeland H, Walhovd KB, Fjell AM. (2023) Hippocampal-cortical functional connectivity during memory encoding and retrieval. Neuroimage 279:120309. doi: 10.1016/j.neuroimage.2023.120309

[R5] Lintas A, Mesrobian SK, Bader M, Villa AEP. (2021) ERPs in controls and ADHD patients during dual n-back task. Advances in cognitive neurodynamics (VII), pp. 189–203. Springer Nature Singapore. doi: 10.1007/978-981-16-0317-4_20

[R6] Rugg MD, Vilberg KL. (2013) Brain networks underlying episodic memory retrieval. Curr Opin Neurobiol. 23(2):255-60. doi: 10.1016/j.conb.

Round 2

Reviewer 2 Report (New Reviewer)

Comments and Suggestions for Authors

The revision is satisfactory.

This manuscript is a resubmission of an earlier submission. The following is a list of the peer review reports and author responses from that submission.

Round 1

Reviewer 1 Report

Comments and Suggestions for Authors

The manuscript is generally well-written but some points can improve the quality of the work:

1- Abstract: the background and aim is unclear

2- Introduction: it is better to start with limitations of previous studies and missing gaps and then put the aim of the work: it seems confusing to write one study limitation and add an individual aim or hypothesis!

3- all figures miss signs of significance

4- Any sex difference?

Comments on the Quality of English Language

minor errors

Author Response

Reviewer 1

1- Abstract: the background and aim is unclear

We apologized for the lack of clarity in the original abstract, and we thank the reviewer for this comment. We have rewritten the Abstract to make it clearer.

“ABSTRACT: Pattern separation is considered a crucial process that allows us to distinguish among the highly similar and overlapping experiences which constitute our episodic memory.  Not only different episodes share common features, but it is often the case that they share the context in which they occurred. While there has been a great number of studies investigating pattern separation, and its behavioral counterpart, a process known as mnemonic discrimination, surprisingly, research exploring the influence of context on pattern separation or mnemonic discrimination has been less common. The available evidence showed that similar items with similar context led to a failure in pattern separation due to high similarity that triggers overlap between events. On the other hand, others have shown that pattern separation can take place even under these conditions, allowing humans to distinguish between events with similar items and contexts, as different hippocampal subfields would play complementary roles in enabling both pattern separation and pattern completion. In the present study, we were interested in testing how stability in context influenced pattern separation. Despite the fact that pattern separation is by definition an encoding computation the existing literature has focused on the retrieval phase. Here, we used a subsequent memory paradigm in which we manipulated the similarity of context during encoding of visual objects selected from diverse categories. Thus, we manipulated the encoded context of each object category (four items within a category), so that some categories had the same context and others had different context. These allowed us to test not only the items presented, but also include the conditions that entail the greatest demand on pattern separation. After a 20-minute period, participants performed a visual mnemonic discrimination task in which they had to differentiate between old, similar, and new items by providing one of the three options for each tested item. Similarly to previous studies, we found no interaction between judgments and contexts, and participants were able to discriminate between old and lure items at the behavioural level in both conditions. Moreover, when averaging the ERPs of all the items presented within a category, a significant SME emerged between hits and new misses, but not between hits and old false alarms or similar false alarms. These results suggest that item recognition emerges from the interaction with subsequently encoded information, and not just between item memory strength and retrieval processes.”

2- Introduction: it is better to start with limitations of previous studies and missing gaps and then put the aim of the work: it seems confusing to write one study limitation and add an individual aim or hypothesis!

We appreciate your recommendations to improve the introduction. We have revised it to first explain the limitations of previous studies before stating the aim of our study. Additionally, we outlined our approach to address each of these limitations. We also followed the comments of the editors and the other reviewers in rewriting the introduction. We hope it is more cohesive and clearer after these changes.

“Episodic memory refers to the ability to form and retrieve memories of specific past events [1]. This simple description, however, hides a highly complex process. Human beings encode an immense amount of life episodes, each of them associated with a spatial and temporal moment [2], which constitute the context. Commonly, context refers to the general information that is associated with a specific event at the time of encoding [3]. The use of this contextual information is a defining feature of episodic memory [1,4-6]. Although each event is unique, it is often the case that it shares common features with previous episodes, and even shares the context in which they occurred. When we encode similar information, there is a risk of interference between memories due to overlapping neural representations [7-8]. How, then, can humans distinguish among these memories? Our brain uses a process known as pattern separation to form unique representations for similar experiences, allowing us to recollect specific details [9-14]. Pattern separation, thus, implies the capacity to resolve interference from an overlap in stimulus features and associated neural responses [15].

In the last decade there has been a great number of studies investigating the neural correlates of pattern separation, and its behavioral counterpart, a process known as mnemonic discrimination [16-17]. Surprisingly, research exploring the influence of context on pattern separation or mnemonic discrimination has been less common [18-25]. Thus, differentially from previous pattern separation research, these studies used the integration of pictured objects in contexts. Although with certain variations, they commonly presented images of objects in the background of a unique scene at encoding. Interestingly, they showed that background context facilitated target recognition but also increased the rate of false recognition of lure similar items [19,23-25], reflecting a reduced mnemonic discrimination. Crucially, Libby, Reagh, Bouffard, Ragland, and Ranganath [22] found that hippocampal activity generalized across similar objects that were encoded in the same context [see also 20].

In the present study we were interested in investigating the effect of context on mnemonic discrimination and its neural counterpart (i.e. pattern separation). Several aspects differentiate our study from previous research. First, even though pattern separation is by definition an encoding computation [26], the above mentioned studies have focused on the retrieval phase. Second, several studies have shown that interference between memory traces is frequently accounted by their associations to similar contexts [20]. Thus, differentially from previous research, we explored whether successful target identification and mnemonic discrimination can be predicted by ERPs during encoding [21,27-28], when the object-context binding is formed [29]. Finally, previous research has demonstrated that mnemonic discrimination decreased when studying an increased number of related items [30-31]. Thus, visual objects were selected from diverse categories, and, in contrast to previous studies, for each of these categories four exemplars were presented. With this aim, we used EEG to register the neural activity during a mnemonic discrimination task [10,33], and applied a subsequent memory approach.

Since most of the previous studies have used fMRI, very little is known about the temporal dynamics of pattern separation [21,28,33]. Thus, by recording ERPs during our mnemonic discrimination task we can increase the understanding of the temporal dimension of pattern separation. Furthermore, as recently stated by Amer and Davachi [15], there is an increased interest in the contribution of extra-hippocampal regions to the process of pattern separation, which is best considered as a process supported by a network of brain regions (15,34-36]. Finally, ERPs are considered a valuable tool to investigate the neural activity at encoding in the subsequent memory paradigm due to fast and brief nature of neurocognitive processes allowing to separate different subprocesses [37].

Since we were interested in studying the influence of contextual stability at encoding [38], in this task, participants learned visual objects from a category which were presented either on the same or on a different background context. Following previous studies [38-39], in the recognition phase visual objects were presented without their correspondent background context [23,29] and participants had to differentiate between old (studied object), similar (new items from a previously seen category) and new items (new objects from a new category). Considering the available evidence, we predict that accuracy for target identification will increase for objects presented in the same background context relative to objects presented in different contexts. On the other hand, correct identification of a similar lure (i.e. mnemonic discrimination) will decrease for objects presented in the same background context relative to objects presented in different contexts, because they match both object and context information to a greater degree [30,40].  This can be explained by a generalization process that occurs at the expense of detailed memory for those objects [41]. Regarding the ERP correlates, a recent review [37] proposed a functional organization of ERP SMEs into three main components: two early frontal and parietal components starting at about 300ms after stimulus onset, reflecting semantic processing and the binding of multiple features of the event, respectively; and a third component, described as a sustained late frontal (starting at around 550 ms) reflecting associative and conceptual encoding. Previous studies have also suggested that mnemonic operation promoting later memorability can be divided into subprocesses, one starting at 300ms and the other at around 500ms [42-44]. Accordingly, we hypothesized that correct target identification would be related to a greater positive amplitude starting at around 300 and around 500ms in frontal and centroparietal electrodes at encoding phase. Similar lures incorrectly identified as old objects will exhibit a greater positive amplitude in frontal and centroparietal electrodes at encoding phase, similar to the increase of correct recalls [20-21,31]. We also predict that objects encoded in the same context will show greater amplitude than those encoded in different contexts.”

3- all figures miss signs of significance

We apologize for the confusion regarding the significance in the figures. Significance differences are indicated by an asterisk in the ERPs, with a line connecting the conditions where significance is observed, and by black circles in the topographic figures. These details are provided in the figure captions for further reference. We hope this clarification helps in interpreting the data presented. We also included tables 1 and 2, along with their appendices, to reflect the differences in significance in the figures 2 and 4.

“Figure 5. Encoding, Category analysis (Response). Significant differences are marked with an asterisk in the ERPs and with black circles in the topography. Left: ERPs from “old hit” (blue line), “similar FA” (red line), “similar CR” (green line), “old FA”, and “new miss” in the 500-800 ms time window. Right: topographic map of the grand-average of “old hits-new miss” in the 500-800 ms time window.”

“Figure 6. Encoding, Category analysis (Context). Significant differences are marked with an asterisk in the ERPs and with black circles in the topography. Top and Left: ERPs from “same context” (grey line), and “different context” (black line) in the 300-500 ms time window. Bottom and Left: topographic map of the grand-average of “same context-different context” in the 300-500 ms time window. Top and Right: ERPs from “same context” (grey line), and “different context” (black line) in the 500-800 ms time window. Bottom and Right: topographic map of the grand-average of “same context-different context” in the 500-800 ms time window.”

“Figure 7. Encoding, Last Item analysis (Context). Significant differences are marked with an asterisk in the ERPs and with black circles in the topography. Top and Left: ERPs from “same context” (grey line), and “different context” (black line) in the 300-500 ms time window. Bottom and Left: topographic map of the grand-average of “same context-different context” in the 300-500 ms time window. Top and Right: ERPs from “same context” (grey line), and “different context” (black line) in the 500-800 ms time window. Bottom and Right: topographic map of the grand-average of “same context-different context” in the 500-800 ms time window.”

Table 1

Table Mean proportion of responses

M

SD

Significant differences*

Old response to Target (1)

42.900

2.947

2,3,6,7,9

Old response to Lure (2)

22.209

2.245

1,3,5,9

Old response to Foil (3)

1.887

.473

1,2,4,5,6,7,8,9

Similar response to Target (4)

36.167

3.093

3,5,6,9

Similar response to Lure (5)

48.674

3.004

2,3,4,6,7,8,9

Similar response to Foil (6)

16.930

1.838

1,3,4,5,9

New response to Target (7)

20.933

2.258

1,3,5,8,9

New response to Lure (8)

29.117

2.692

3,5,7,9

New response to Foil (9)

81.183

2.035

1,2,3,4,5,6,7,8

*See appendix 1 for pairwise comparisons

Table 2

Reaction time responses

M

SD

Significant differences*

Old response to Target (1)

0,962

0,083

4,5

Old response to Lure (2)

0,986

0,084

4

Similar response to Target (3)

1,030

0,087

6,4,7,8

Similar response to Lure (4)

1,054

0,079

1,3,6,7,8

Similar response to Foil (5)

1,040

0,099

1,6,7,8

New response to Target (6)

0,955

0,114

3,4,5

New response to Lure (7)

0,955

0,114

3,4,5

New response to Foil (8)

0,933

0,100

1,6,7,8

*See appendix 2 for pairwise comparisons

4- Any sex difference?

We thank you for your question regarding potential sex differences in our data.  To address this, we conducted a mixed-design ANOVA with “response” as the within-subjects factor and “sex” as the between-subjects factor. This analysis allowed us to evaluate both the main effect of response type and its interaction with sex to determine if response patterns differed between male and female participants. The results indicated a significant main effect of response, (p<.05), but the interaction between response and sex was not statistically significant (p>.05), indicating that response patterns did not differ significantly between male and female participants. It is important to note that our sample included 20 female and 6 male participants. Given this imbalance, we interpret these results with caution, as this sample size disparity may affect the interpretation of any between-group comparisons. These results can be consulted in the analysis tables presented below this answer. We hope this clarifies the approach and results.

sex

res

Mean

Standard error

95% Confidence interval

Lower limit

Upper limit

male

1

37,430

6,129

24,780

50,080

2

52,770

6,311

39,744

65,796

3

75,180

4,091

66,737

83,624

4

19,446

4,727

9,690

29,203

5

40,224

6,504

26,800

53,648

6

22,346

4,786

12,469

32,223

7

23,406

3,603

15,969

30,842

8

27,784

5,712

15,995

39,573

9

1,414

,998

-,646

3,473

female

1

44,541

3,357

37,612

51,470

2

47,446

3,457

40,311

54,581

3

82,984

2,241

78,359

87,609

4

23,037

2,589

17,694

28,381

5

34,950

3,562

27,598

42,303

6

20,509

2,621

15,099

25,919

7

14,987

1,974

10,914

19,060

8

29,517

3,129

23,060

35,974

9

2,029

,547

,901

3,157

Within-subjects effects

Origin

Sum of squares type III

gl

Quadratic mean

F

Sig.

res

Sphericity assumed

73108,224

8

9138,528

53,372

,000

Greenhouse-Geisser

73108,224

2,535

28839,527

53,372

,000

Huynh-Feldt

73108,224

2,977

24554,333

53,372

,000

Lower-limit

73108,224

1,000

73108,224

53,372

,000

res * sex

Sphericity assumed

1191,408

8

148,926

,870

,543

Greenhouse-Geisser

1191,408

2,535

469,983

,870

,446

Huynh-Feldt

1191,408

2,977

400,150

,870

,460

Lower-limit

1191,408

1,000

1191,408

,870

,360

Error(res)

Sphericity assumed

32874,684

192

171,222

Greenhouse-Geisser

32874,684

60,840

540,346

Huynh-Feldt

32874,684

71,458

460,058

Lower-limit

32874,684

24,000

1369,779

Reviewer 2 Report

Comments and Suggestions for Authors

In the submitted paper, the authors analyze the incidence of the context in the ability to differentiate similar and different objects through the process. They focus on this effect during the encoding phase and test it after a delay of 20 minutes. Additionally, they record electrophysiological data to evaluate the link between the observed ERPs and the type of object the subject is trying to identify. The paper is interesting, as there is a lack of memory tasks that focus on object discrimination and also little information regarding the neural basis of this process. However, I have some concerns regarding the analysis and presentation of the data

Major issues:

1.     1.1 In the Methods section, the presentation of different backgrounds during encoding is described like this:
A category from the different context group has also four different category objects, but each one of them with a different background. In addition, backgrounds are not repeated across categories.”
1.2 I wonder if the differences between backgrounds were parametrized in some way to ensure that the difference between them is effectively homogeneous and not some more alike than others.

1.3  Additionally, would be worth to know if the backgrounds used in the group “different context” were used in the “same context” group and if the backgrounds used in this last group were the same or they changed between categories.

2      2.1 The results section is organized in a way that makes it very difficult to read. While the inclusion of test and descriptive statistics for each paired comparison is desirable, it could be more easily understood if reorganized in a table.
2.2 It would be also important to have individual data included in figure 2 graphics in order to see how they are distributed and have a more complete picture of the results. 2.3 Having in mind that some significant differences were observed in reaction times, it would be important to add this information into a graph.

2.4  Finally, in figure 3, New Hit data is not shown.

3      3.1 Given the following statement from discussion:
Furthermore, ERPs from the encoding of image categories that were subsequently recognized (old hit) were significantly different from those categories that were subsequently marked as new (new miss indicates both category and item non-recognition). These results suggest a category recollection in which the consecutive presentation of similar items within a category created a strong category-related memory trace (Herz et al., 2023; Wing et al., 2020). This trace is reflected in an increased ERP positivity of those categories which were previously seen and recognized in comparison to non-recognized and marked-as-new items, as the last one implied less memory encoding strengthening and consequently a weaker trace “
Having the New Hit data in the graphic becomes relevant since a similar effect (as New Miss) in its ERP should be observed if the category-related memory trace is the cause of a smaller amplitude.

Minor concerns:

1.     In order to understand how the change of background could affect the encoding process, it would be worth to know the full instructions given to the participants before the beginning of the task. In particular, whether they were warned that background would not be evaluated in the test phase so they would know how much attention to pay to it. Or if, on the contrary, nothing was said so that they might have paid attention to the whole picture because the whole scene would be evaluated.

2.     As authors said, at the test phase “Participants responded while the picture is on screen by pressing key number 1 if it was an old item, key number 2 if it was a similar item or key number 3 if it was a new item.” Since reaction time was measured, it would be important to know if these reference were showed in the screen at the moment of the answer or if the participant needed to remember which number corresponded to which type of object.

3.     An untimed break was taken by subjects during the test phase (“There was one untimed break in the middle of the discrimination phase”). The justification of this break and why it was untimed are not mentioned.

4.     The sample size was calculated based in an effect size of 0,25 (“Sample size was calculated using G*Power (Faul, Erdfelder, Lang & Buchner, 2007). With the aim of detecting an effect size of 0.25 and obtaining a statistical power of 0.95 the required sample was determined to be at least of 18 participants.”) but the authors do not mention where that values comes from.

5.      In the results section, consistency of graphs could be improved. It would be clearer if the color code had a logical relationship between them. Similarly, since in figure 2 panels A and B exhibit the same magnitude (mean proportion of responses), it would be clearer for reading if they had the same scale on the Y axis. It would also be very helpful to indicate significant differences within the graph.

6.     Finally, in the discussion a consideration is made regarding the reaction times with reference to results from the same authors. “Moreover, old false alarms were slower than old hits and new CR. These results are consistent with previous literature, such as García-Rueda et al. (2019), who found significantly slower reaction times for similar CR in comparison with old hits, similar FA and new CR”. It would be interesting to compare reaction times  with results from other authors that used a similar task, as the background could add a new level of processing that may interact with the similarity of the items.

Comments on the Quality of English Language

The English is mostly correct

Author Response

Reviewer 2

Major issues:

  1. 1.1 In the Methods section, the presentation of different backgrounds during encoding is described like this:
    A category from the different context group has also four different category objects, but each one of them with a different background. In addition, backgrounds are not repeated across categories.”
    1.2 I wonder if the differences between backgrounds were parametrized in some way to ensure that the difference between them is effectively homogeneous and not some more alike than others.

We appreciate the question regarding the presentation of different backgrounds during the encoding phase. Although we did not employ a formal parametrization method to quantify the differences between backgrounds, they were randomly assigned to the “same context” and “different context” conditions for each participant. Thus, the list of backgrounds that were repeated (“same context”) or not (“different context”) were unique to each participant. This randomization helps to minimize potential biases related to the distinctiveness or other characteristics of the backgrounds.  

1.3 Additionally, would be worth to know if the backgrounds used in the group “different context” were used in the “same context” group and if the backgrounds used in this last group were the same or they changed between categories.

We appreciate the question about the background conditions and would like to clarify that the backgrounds assigned to the “same context” condition were not simultaneously assigned to the “different context “condition. In other words, the backgrounds that were repeated for a particular object category (“same context” condition) were only presented within that specific object category. In addition, the backgrounds assigned to each object category are always unique between object categories, i.e. the background chosen to be repeated within one object category does not appear in any other object category belonging to the “same context” condition. However, a random list of backgrounds was generated for each participant, allowing for the possibility that one participant may have viewed a background in the “same context” condition while another participant viewed that specific background as part of the “different context” condition.

2      2.1 The results section is organized in a way that makes it very difficult to read. While the inclusion of test and descriptive statistics for each paired comparison is desirable, it could be more easily understood if reorganized in a table.

We apologize for the lack of clarity in our results and appreciate your comment. We have added a column in table 1 reflecting the significant differences between responses and an appendix presenting the pairwise comparisons to more easily reflect our behavioural results. We hope this table will improve the quality of our manuscript.

Appendix 1:

Pairwise comparisons

(I)Response

(J)Response

Mean differences (I-J)

Sig.b

Old response to target

New response to target

21,967*

,001

Old response to lure

20,691*

,000

New response to foil

-38,283*

,000

Similar response to foil

25,970*

,000

Old response to foil

41,013*

,000

Similar response to target

Similar response to lure

-12,507*

,000

New response to foil

-45,016*

,000

Similar response to foil

19,237*

,000

Old response to foil

34,280*

,000

New response to target

Similar response to lure

-27,742*

,000

New response to lure

-8,184*

,000

New response to foil

-60,250*

,000

Old response to foil

19,046*

,000

Similar response to lure

Old response to lure

26,466*

,000

New response to lure

19,558*

,036

New response to foil

-32,509*

,000

Similar response to foil

31,745*

,000

Old response to foil

46,787*

,000

Old response to lure

New response to foil

-58,974*

,000

Old response to foil

20,322*

,000

New response to lure

New response to foil

-52,066*

,000

Old response to foil

27,230*

,000

New response to foil

Similar response to foil

64,253*

,000

Old response to foil

79,296*

,000

Similar response to foil

Old response to foil

15,043*

,000

b. Adjustment for multiple comparisons: Bonferroni.

2.2 It would be also important to have individual data included in figure 2 graphics in order to see how they are distributed and have a more complete picture of the results.

We appreciate your suggestion. In response, we have included box-and-whisker plots in the behavioral section. These figures represent the range and distribution of individual scores for each response type. The plots provide a clearer visualization of the data distribution, supporting a more comprehensive picture of the results.

Figure 2. A. Mean proportion of old, similar and new responses to targets, lures and foils. B. Mean proportion of old, similar and new responses to targets from same or different context and lures from same or different context.

2.3 Having in mind that some significant differences were observed in reaction times, it would be important to add this information into a graph.

We appreciate your suggestion. In response, we have included a box-and-whisker plot in the behavioral section along with a table (table 2) where the significant differences can be observed. Additionally, we included an appendix where the pairwise comparisons can be evaluated in more detail.

Figure 4. Reaction time responses of old, similar and new responses to targets, lures and foils.

Table 2

Reaction time responses

M

SD

Significant differences*

Old response to Target (1)

0,962

0,083

4,5

Old response to Lure (2)

0,986

0,084

4

Similar response to Target (3)

1,030

0,087

6,4,7,8

Similar response to Lure (4)

1,054

0,079

1,3,6,7,8

Similar response to Foil (5)

1,040

0,099

1,6,7,8

New response to Target (6)

0,955

0,114

3,4,5

New response to Lure (7)

0,955

0,114

3,4,5

New response to Foil (8)

0,933

0,100

1,6,7,8

*See appendix 2 for pairwise comparisons

Appendix 2:

Pairwise comparisons

(I)Response

(J)Response

Mean differences (I-J)

Sig.b

Old response to target

Similar response to lure

-,092*

,002

Similar response to foil

-,078*

,011

Similar response to target

New response to target

,075*

,003

Similar response to lure

-,024*

,004

New response to lure

,075*

,015

New response to foil

,097*

,000

New response to target

Similar response to lure

-,099*

,000

Similar response to foil

-,085*

,000

Similar response to lure

Old response to lure

,068*

,041

New response to lure

,100*

,001

New response to foil

,121*

,000

New response to lure

Similar response to foil

-,086*

,001

New response to foil

Similar response to foil

-,107*

,000

b. Adjustment for multiple comparisons: Bonferroni.

2.4  Finally, in figure 3, New Hit data is not shown.

 We appreciate your comment and would like to clarify that Figure 3 shows the ERPs of the encoding phase. The response conditions shown in this image are as follows: old hits (responding “old” to an object that was presented), similar FA (responding “old” to a new object but from a category that was presented), similar CR (responding “similar” to a new object from a category that was presented), old FA (responding “similar” to an object that was presented), and new miss (responding “new” to an object that was presented). The reason why new hits are not included is that they correspond to new responses to new objects from categories that had not been previously seen (otherwise, they will be new responses to lures). Thus, their EEG signal could not be recorded during the encoding phase. For this reason, only the conditions with previously seen objects and/or categories are included in figure 3. Nevertheless, new hits provide valuable information and are analyzed in the behavioural section.

3      3.1 Given the following statement from discussion:
Furthermore, ERPs from the encoding of image categories that were subsequently recognized (old hit) were significantly different from those categories that were subsequently marked as new (new miss indicates both category and item non-recognition). These results suggest a category recollection in which the consecutive presentation of similar items within a category created a strong category-related memory trace (Herz et al., 2023; Wing et al., 2020). This trace is reflected in an increased ERP positivity of those categories which were previously seen and recognized in comparison to non-recognized and marked-as-new items, as the last one implied less memory encoding strengthening and consequently a weaker trace “
Having the New Hit data in the graphic becomes relevant since a similar effect (as New Miss) in its ERP should be observed if the category-related memory trace is the cause of a smaller amplitude.

For the reasons described in the previous answer, it was not possible to make this comparison in ERPs. The “new miss” condition implies that participants do not recognize the object or category of an image presented during encoding. In contrast, “new hits” correspond to images that participants correctly identify as new, meaning their presentation could not have been recorded during encoding.

Minor concerns:

  1. In order to understand how the change of background could affect the encoding process, it would be worth to know the full instructions given to the participants before the beginning of the task. In particular, whether they were warned that background would not be evaluated in the test phase so they would know how much attention to pay to it. Or if, on the contrary, nothing was said so that they might have paid attention to the whole picture because the whole scene would be evaluated.

It was explained to the participants that they should pay attention to the whole picture, but that they would only be asked about the objects that appear in them. It was also explained to them that the images would be presented in four blocks between which they would have untimed breaks. When they finished a block, an instruction appeared on the screen indicating that they were in a pause and that to continue they only had to press the “space bar” on the keyboard. We have modified this information in the methodology to specifically address this point.

“The stimuli and procedure used in this study were adapted from the methods described by Poch, Prieto, Hinojosa, and Campo [12]. A total of 1280 images were shown to participants during the study phase. In the instructions provided to participants, they were asked to pay attention to all the images, but that they would only be asked about the objects that appear in them. These images represented objects (without human body parts or animals) from different categories, and each category contained four images (see Figure 1). The categories were divided into two groups: same context (160 categories) or different context (160 categories). A category from the same context group has four different category objects (a total of 640 images), but all of them have the same background. A category from the different context group has also four different category objects (a total of 640 images), but each one of them with a different background. In addition, backgrounds are not repeated across categories, i.e. each category has a unique background (or backgrounds in the case of belonging to the "different context" group). The presentation of each image lasted for 1500 ms, and a 1000 ms grey screen was displayed after each image. In addition, the images were presented in four blocks of 320 images each, allowing participants to take untimed breaks between blocks. Participants had a 20-minute break after completing the study phase, and then moved on to the discrimination phase. 

In the discrimination phase, 400 images were randomly presented to each participant. In line with the methods by Poch, Prieto, Hinojosa, and Campo [12], we tested the first object presented within each category to ensure that the effects were due to stimulus interference. These images were divided into 160 old items (previously presented in the study phase), 160 lure items (new images, but from a previously seen object category), and 80 new items (new images from new object categories). Furthermore, the old and similar items were divided into two groups: category encoded with the same context or category encoded with different contexts. Thus, there were 80 old items from the encoded categories with the same context, 80 old items from the encoded categories with different contexts, 80 lure items from the encoded categories with same context, 80 lure items from the encoded categories with different context, and finally 80 new items. All images were presented without background for 1500 ms each and followed by a 2000 ms white screen. Participants were instructed to press key numbers 1, 2, or 3 while the image is on the screen to indicate whether the items were old, lure, or new, respectively. Participants were given a non-timed pause at the midpoint of the discrimination phase, after the first 200 images.”

  1. As authors said, at the test phase “Participants responded while the picture is on screen by pressing key number 1 if it was an old item, key number 2 if it was a similar item or key number 3 if it was a new item.” Since reaction time was measured, it would be important to know if these reference were showed in the screen at the moment of the answer or if the participant needed to remember which number corresponded to which type of object.

We appreciate your question. Participants were verbally instructed about the correspondence between each number and response type. Additionally, these instructions were displayed on the screen before starting the test. During the trials, these response options (numbers) were not displayed on the screen, but during each break, the instructions were shown again on the screen to remind participants of the correct key-response associations, so they did not have to rely on memory during the break. If the reviewer considers it necessary, we can add this information to the methodology section or discuss it as a limitation of the study.

  1. An untimed break was taken by subjects during the test phase (“There was one untimed break in the middle of the discrimination phase”). The justification of this break and why it was untimed are not mentioned.

Participants were given a non-timed pause at the midpoint of the discrimination phase, after the first 200 images, to reduce fatigue and maintain focus. The break was untimed to allow participants to rest adequately and resume when they felt ready, ensuring that performance in the second half of the task was not compromised by fatigue or time pressure. The test phase lasted approximately 24 minutes, so it was divided into two blocks of 12 minutes each. During the encoding phase, the same criterion was followed by dividing the study into blocks of 320 pictures, so each block lasted approximately 13 minutes. We took into account previous studies that suggest the performance decline typically occurs within the first 15 minutes (some references below). If the reviewer considers it necessary, we can add this information to the methodology section for additional clarity.

  • Frost, H. G. (1965, January). Observations on a great occasion. Adult Education, 37, 282–283.
  • Wilson, K., & Korn, J. H. (2007). Attention during lectures: Beyond ten minutes. Teaching of Psychology, 34(2),85- 89.
  • Teichner, W.H. The detection of a simple visual signal as a function of time of watch. Hum. Factors 1974, 16, 339–352.
  • Al-Shargie, F., Tariq, U., Mir, H., Alawar, H., Babiloni, F., & Al-Nashash, H. Vigilance decrement and enhancement techniques: A review. Brain Sci. 2019, 9, 178.
  • Bradbury, N. A. (2016). Attention span during lectures: 8 seconds,10minutes, or more?
  • Hartley, J., & Davies, I. K. (1978). Note-taking: A critical review. Programmed learning and educational technology, 15(3), 207-224.
  • Al-Shargie, F., Tariq, U., Hassanin, O., Mir, H., Babiloni, F, & Al-Nashash, H. Brain Connectivity Analysis Under Semantic Vigilance and Enhanced Mental States. Brain Sci. 2019, 9(12), 3

  1. The sample size was calculated based in an effect size of 0,25 (“Sample size was calculated using G*Power (Faul, Erdfelder, Lang & Buchner, 2007). With the aim of detecting an effect size of 0.25 and obtaining a statistical power of 0.95 the required sample was determined to be at least of 18 participants.”) butthe authors do not mention where that values comes from.

The effect size of 0.25 corresponds to a Cohen’s f value, which is a standard measure used in power analysis for ANOVA designs. It is often used when there is no specific effect size available from prior studies and serves as a reasonable estimate. We included this value in our sample size calculation in order to ensure sufficient power (0.95) for detecting significant effects.

  1. In the results section, consistency of graphs could be improved. It would be clearer if the color code had a logical relationship between them. Similarly, since in figure 2 panels A and B exhibit the same magnitude (mean proportion of responses), it would be clearer for reading if they had the same scale on the Y axis. It would also be very helpful to indicate significant differences within the graph.

We have not matched color between original figure 2 and 3 because figure 3 only captures responses to targets and lures, since it represents the EEG signal of previously seen objects and/or categories. In contrast, figure 1 also represents responses to new objects from new categories, which precludes matching. Following the suggestions, we have modified the Y-axis in the original figure and changed the color code to match with new behavioural figures. Also, we have added a column in table 1 reflecting the significant differences between responses and an appendix presenting the pairwise comparisons to more easily reflect our behavioural results.

  1. Finally, in the discussion a consideration is made regarding the reaction times with reference to results from the same authors. “Moreover, old false alarms were slower than old hits and new CR. These results are consistent with previous literature, such as García-Rueda et al. (2019), who found significantly slower reaction times for similar CR in comparison with old hits, similar FA and new CR”. It would be interesting to compare reaction times with results from other authors that used a similar task, as the background could add a new level of processing that may interact with the similarity of the items.

We appreciate this suggestion. In response, we have added another study that used contexts within a similar task to examine reactions times. We hope this enhances the discussion’s relevance.

“In addition, reaction time analyses showed that similar CR were significantly slower than old false alarm, old hits, similar FA, and new CR. Moreover, old false alarms were slower than old hits and new CR. These results are consistent with previous literature, such as García-Rueda, Poch, and Campo [10], who found significantly slower reaction times for similar CR in comparison with old hits, similar FA and new CR. This slower reaction time for similar CR suggests that discriminating between items within a category is a more complex process than recognizing previously encoded items or items from an unseen category. Hayes, Nadel, and Ryan [29] similarly found that participants were slowest to respond to the “scene lures” (similar to targets, but novel object presented in a novel scene) than any other condition, supporting the idea that distinguishing related items and scenes involves a more complex processing effort. Moreover, pairwise comparisons between memory judgment (correct or incorrect) and conditions revealed that participants were fasted to respond to “object.object” correct trials (old object presented on a white background during encoding and testing phases), followed by “scene.scene” correct trials (old object presented on an old background). Participants were slowest to respond to the “scene.object” correct trials (an old object without background which was previously presented with a background during study), which did not differ in response times from object lures correct rejections (similar to targets, but novel object on a white background) and scene lures correct rejections. These results are also consistent with our study, as they suggest that participants experienced a greater difficulty discriminating the object when the context is lost, and that this difficulty was comparable to correctly rejecting a lure”.

Reviewer 3 Report

Comments and Suggestions for Authors

This study examines the contextual influences on pattern separation during encoding by employing event-related potentials (ERPs) to explore how background context impacts item memory performance. The main strength of the paper lies in its well-designed experimental paradigm and large trial count, which ensure robust statistical power for each condition. However, there are notable concerns about the study’s theoretical motivation/interpretation and methodology.

Major Comments:

1. Study Motivation and Theoretical Rationale.

The paper lacks a compelling theoretical motivation for examining pattern separation during encoding, specifically using scalp EEG. Traditionally, pattern separation is linked to hippocampal processing, especially within hippocampal subfields (e.g., reviews from Yassa and Stark labs). While there may be potential to observe hippocampal outputs at the scalp level or the cortical contributions to pattern separation, this raises questions about how such processes can be effectively observed with scalp EEG signals. The authors need to clarify the rationale behind studying these processes at the cortical level and explain which cortical regions they hypothesize are involved in pattern separation during encoding. Moreover, given the abundance of literature on the subsequent memory effect ERP research, it is surprising that this body of work was not referenced, which could have strengthened the study’s context and relevance.

2. Justification of Analytical Choices  

The paper lacks clear justification for several analytical choices, including the selection of time windows, ERP components, and the decision to analyze both the first and last items. These decisions should be theoretically grounded to enhance the interpretability and validity of the findings. 

i)      It would be beneficial to explain the basis for choosing the time windows used in the ERP analysis and how these align with known temporal dynamics of memory-encoding processes. I recommend that the author examine the literature on memory-related ERPs closely; this will not only assist the authors in justifying their choice of ERP time windows or components but will also help readers better interpret the principal findings within the broader context of memory ERP research. For instance, the decision to focus on the P300 component without sufficient justification or supporting literature beyond a single master’s thesis (Hollarek, 2015) is concerning. The P300 is well-researched in ERP literature; however, there is no well-established link between this component and pattern separation process. Additional citations and context are necessary to clarify why this component was chosen to measure the temporal dynamics of pattern separation. A stronger integration with relevant P300 research would also benefit the study’s theoretical coherence.

ii)    First and Last Item Analysis: The authors provide procedural explanations for the ‘first item analysis’ and ‘last item analysis’, detailing how these conditions were contrasted. However, it remains unclear why it is important to compare these conditions or to examine the first and last items specifically. It appears the authors aim to investigate brain activity related to integrating multiple similar items with the same or different backgrounds, which could relate to the concept of “pattern separation”. I assume that the first item might be unaffected by interference, while the last item might necessitate pattern separation due to accumulated interference; therefore, successful encoding may require pattern separation. However, there is no direct comparison between the first and last items, and memory outcomes were not factored into the contrast, which limits the study’s ability to directly connect to other subsequent memory effect ERP studies or pattern seperation in genera. 

3. Interpretation of Results and Implications for Pattern Separation

This paper needs a more thorough interpretation of the main findings, especially concerning how they contribute to our understanding of pattern separation and its neural correlates during encoding. The ERP results do not show differences based on behavioral outcomes (hits vs. misses) but rather on distinctions between same and different contexts, regardless of memory success. This approach fails to directly assess the success or failure of pattern separation; instead, it may be capturing conditional differences related to stimulus pairing. Furthermore, the early time window likely reflects the perceptual features of the stimuli rather than any neural process directly linked to pattern separation. This aspect of the analysis limits the interpretation of the findings within the framework of pattern separation and could benefit from more precise analysis to clarify the processes being measured.

Minor comments:

- Please provide a detailed calculation or clear definition of the LDI and explain its relevance to the study’s aims and interpretation. 

- Section 3, behavioral results, currently reads as a summary of analysis results, yet the link between these results and the study’s broader scientific conclusions is ambiguous. For instance, in Lines 302-306, the finding of “more positive activity in a central-anterior cluster of electrodes for old hits compared to new misses” is presented as an observation. However, the inference or conclusion drawn from this observation, along with how it aligns with the study’s objectives, is not clearly articulated. 

Comments on the Quality of English Language

The overall quality of the English language is good. 

Author Response

Major Comments:

  1. Study Motivation and Theoretical Rationale.

The paper lacks a compelling theoretical motivation for examining pattern separation during encoding, specifically using scalp EEG. Traditionally, pattern separation is linked to hippocampal processing, especially within hippocampal subfields (e.g., reviews from Yassa and Stark labs). While there may be potential to observe hippocampal outputs at the scalp level or the cortical contributions to pattern separation, this raises questions about how such processes can be effectively observed with scalp EEG signals. The authors need to clarify the rationale behind studying these processes at the cortical level and explain which cortical regions they hypothesize are involved in pattern separation during encoding. Moreover, given the abundance of literature on the subsequent memory effect ERP research, it is surprising that this body of work was not referenced, which could have strengthened the study’s context and relevance.

We thank the reviewer for raising this point. While most of the research on pattern separation has focused on the role of the hippocampus, in the last few years there has been an increasing interest in understanding the contribution of extra-hippocampal regions to pattern separation. Then, several authors including Dr. Yassa, Dr. Davachi and Dr. Kirwan have explored the neocortical basis of pattern separation, which can be investigated with EEG (Amer & Davachi, 2023; Morcom, 2015; Nash, Hodges, Muncy, & Kirwan, 2021; Pidgeon & Morcom, 2016; Stevenson, Reagh, Chun, Murray, & Yassa, 2020). In addition to this, it is well known that neurocognitive processes are fast and transient and separating different subprocesses requires a good temporal resolution. Because of the poor temporal resolution of the fMRI signal, little is known about the temporal dynamics of pattern separation. Thus, using EEG potentially allows us to identify different subprocesses that can provide better understanding of the temporal dynamics of pattern separation.

We have rewritten the manuscript to clarify the issues pointed by the reviewer.

“In the present study we were interested in investigating the effect of context on mnemonic discrimination and its neural counterpart (i.e. pattern separation). Several aspects differentiate our study from previous research. First, even though pattern separation is by definition an encoding computation (Motley & Kirwan, 2012), the above mentioned studies have focused on the retrieval phase. Second, several studies have shown that interference between memory traces is frequently accounted by their associations to similar contexts (Herz et al., 2023). Thus, differentially from previous research, we explored whether successful target identification and mnemonic discrimination can be predicted by ERPs during encoding (García-Rueda, Poch, & Campo, 2024; Rollins, Khuu, & Bennett, 2024), when the object-context binding is formed (Hayes, Nadel, & Ryan, 2007). Finally, previous research has demonstrated that mnemonic discrimination decreased when studying an increased number of related items (Arndt, 2010; Wing et al., 2020). Thus, visual objects were selected from diverse categories, and, in contrast to previous studies, for each of these categories four exemplars were presented. With this aim, we used EEG to register the neural activity during a mnemonic discrimination task (Comino Garcia-Munoz et al., 2023; García-Rueda, Poch, & Campo, 2022), and applied a subsequent memory approach.

Since most of the previous studies has used fMRI, very little is known about the temporal dynamics of pattern separation (Hollarek, 2015; Morcom, 2015; Rollins et al., 2024). Thus, by recording ERPs during our mnemonic discrimination task we can increase the understanding of the temporal dimension of pattern separation. Furthermore, as recently stated by Amer and Davachi (Amer & Davachi, 2023), there is an increased interest in the contribution of extra-hippocampal regions to the process of pattern separation, which is best considered as a process supported by a network of brain regions (Amer & Davachi, 2023; Nash, Hodges, Muncy, & Kirwan, 2021; Pidgeon & Morcom, 2016; Stevenson, Reagh, Chun, Murray, & Yassa, 2020). Finally, ERPs are considered a valuable tool to investigate the neural activity at encoding in the subsequent memory paradigm due to fast and brief nature of neurocognitive processes allowing to separate different subprocesses (Mecklinger & Kamp, 2023).”

“Regarding the ERP correlates, a recent review (Mecklinger & Kamp, 2023) proposed a functional organization of ERP SMEs into three main components: two early frontal and parietal components starting at about 300ms after stimulus onset, reflecting semantic processing and the binding of multiple features of the event, respectively; and a third component, described as a sustained late frontal (starting at around 550 ms) reflecting associative and conceptual encoding. Previous studies have also suggested that mnemonic operation promoting later memorability can be divided into subprocesses, one starting at 300ms and the other at around 500ms (Fell, Klaver, Elger, & Fernandez, 2002; Fell et al., 2001; Fernandez et al., 1999). Accordingly, we hypothesized that correct target identification would be related to a greater positive amplitude starting at around 300 and around 500ms in frontal and centroparietal electrodes at encoding phase.”

Amer, T., & Davachi, L. (2023). Extra-hippocampal contributions to pattern separation. Elife, 12. doi:10.7554/eLife.82250

Morcom, A. M. (2015). Resisting false recognition: An ERP study of lure discrimination. Brain Res, 1624, 336-348. doi:10.1016/j.brainres.2015.07.049

Nash, M. I., Hodges, C. B., Muncy, N. M., & Kirwan, C. B. (2021). Pattern separation beyond the hippocampus: A high-resolution whole-brain investigation of mnemonic discrimination in healthy adults. Hippocampus, 31(4), 408-421. doi:10.1002/hipo.23299

Pidgeon, L. M., & Morcom, A. M. (2016). Cortical pattern separation and item-specific memory encoding. Neuropsychologia, 85, 256-271. doi:10.1016/j.neuropsychologia.2016.03.026

Stevenson, R. F., Reagh, Z. M., Chun, A. P., Murray, E. A., & Yassa, M. A. (2020). Pattern Separation and Source Memory Engage Distinct Hippocampal and Neocortical Regions during Retrieval. J Neurosci, 40(4), 843-851. doi:10.1523/JNEUROSCI.0564-19.2019

  1. Justification of Analytical Choices  

The paper lacks clear justification for several analytical choices, including the selection of time windows, ERP components, and the decision to analyze both the first and last items. These decisions should be theoretically grounded to enhance the interpretability and validity of the findings. 

  1. It would be beneficial to explain the basis for choosing the time windows used in the ERP analysis and how these align with known temporal dynamics of memory-encoding processes. I recommend that the author examine the literature on memory-related ERPs closely; this will not only assist the authors in justifying their choice of ERP time windows or components but will also help readers better interpret the principal findings within the broader context of memory ERP research. For instance, the decision to focus on the P300 component without sufficient justification or supporting literature beyond a single master’s thesis (Hollarek, 2015) is concerning. The P300 is well-researched in ERP literature; however, there is no well-established link between this component and pattern separation process. Additional citations and context are necessary to clarify why this component was chosen to measure the temporal dynamics of pattern separation. A stronger integration with relevant P300 research would also benefit the study’s theoretical coherence.

We thank the reviewer for this comment. We have now referred to a recent review of ERPs associated with subsequent memory effect (Mecklinger & Kamp, 2023), and removed any reference to the P300.

“Regarding the ERP correlates, a recent review (Mecklinger & Kamp, 2023) showed that SME are associated with two early frontal and parietal components (300 – 600ms) reflecting semantic processing and the binding of multiple features of the event, respectively. Accordingly, we hypothesized that correct target identification would be related with a greater positive amplitude in the 300-600ms time window in frontal and centroparietal electrodes at encoding phase. Similar lures incorrectly identified as old objects will exhibit a greater positive amplitude in frontal and centroparietal electrodes at encoding phase, similar to the increase of correct recalls (Herz et al., 2023; Hollarek, 2015; Wing et al., 2020). We also predict that objects encoded in the same context will show greater amplitude than those encoded in different contexts.”

  1. First and Last Item Analysis: The authors provide procedural explanations for the ‘first item analysis’ and ‘last item analysis’, detailing how these conditions were contrasted. However, it remains unclear why it is important to compare these conditions or to examine the first and last items specifically. It appears the authors aim to investigate brain activity related to integrating multiple similar items with the same or different backgrounds, which could relate to the concept of “pattern separation”. I assume that the first item might be unaffected by interference, while the last item might necessitate pattern separation due to accumulated interference; therefore, successful encoding may require pattern separation. However, there is no direct comparison between the first and last items, and memory outcomes were not factored into the contrast, which limits the study’s ability to directly connect to other subsequent memory effect ERP studies or pattern separation in general. 

We thank the reviewer for this comment and would like to clarify that our first and last item analyses take memory outcomes into account. Specifically, in the first item analysis, we included old hit, old false alarm, and new miss, as they correspond to items that were subsequently tested (targets). This allows us to examine the subsequent memory effect. For the last item analysis, we included old hit, old false alarm, new miss, and also similar false alarm and similar correct rejections. In this analysis, we specifically examine the fourth presentation of each object category, where participants have either seen the same context four times or four different contexts, depending on whether the object category belonged to the same or different context condition. This allows us to examine how stability of context influence pattern separation during encoding.

The reviewer is right in noting that the first presented item of each category is unaffected by interference, whereas the last item is affected by interference at encoding. Following Reviewer’s suggestion, we compared the first and last items. Our analysis did not reveal any significant cluster when comparing the first and last presentations. Specifically:

- For old hits, no clusters were found when comparing the first and last presentation.

- For similar miss, we found nine positive clusters in the 300-500 ms time window and three positive clusters in the 500-800 ms time window, however, none of them reached significance.

- For new miss, we found one positive cluster in the early time window, which was also no significant.

If the reviewer requires further clarification or additional analyses, please let us know.

  1. Interpretation of Results and Implications for Pattern Separation

This paper needs a more thorough interpretation of the main findings, especially concerning how they contribute to our understanding of pattern separation and its neural correlates during encoding. The ERP results do not show differences based on behavioral outcomes (hits vs. misses) but rather on distinctions between same and different contexts, regardless of memory success. This approach fails to directly assess the success or failure of pattern separation; instead, it may be capturing conditional differences related to stimulus pairing. Furthermore, the early time window likely reflects the perceptual features of the stimuli rather than any neural process directly linked to pattern separation. This aspect of the analysis limits the interpretation of the findings within the framework of pattern separation and could benefit from more precise analysis to clarify the processes being measured.

We thank the reviewer for this comment. Since this is the first study exploring the temporal dynamics of contextual stability in pattern separation, we do not have many references to compare with. Nonetheless, we have discussed our results in light of previous related studies. We found effects in two different time windows, which matched with current literature of SME effects (see Mecklinger & Kamp, 2023). We would like to ask the reviewer more details about the analysis he/she has in mind.

We would like to clarify that in figure 5, the ERP responses during encoding are presented based on subsequent memory judgments. In this figure, we observe a significant difference between old hits and new misses, indicating that successful recognition of the presented category objects differs from those categories that are not subsequently recognized (this is, SME). Specifically, old hits showed a more positive amplitude in the late time window (500-800 ms) compared to new misses.

Additionally, as the reviewer noted, differences based on context are also observed (figures 6 and 7) in both early and late time windows, independent of memory judgment. Thus, although we did not find an interaction between context and memory judgments, we did observe a main effect for both factors.

Minor comments:

- Please provide a detailed calculation or clear definition of the LDI and explain its relevance to the study’s aims and interpretation. 

In the “statistical analysis; behavioral data” section the LDI is described as follows:

“We calculated a lure discrimination index (LDI) in terms of context, which was defined as the ability to rejects similar lures and was calculated as the proportion of correctly identified lures corrected for the baseline rate of similar responses to novel items [33].”

  1. Morcom, A. M. (2015). Resisting false recognition: An ERP study of lure discrimination. Brain Research, 1624, 336–348. https://doi.org/ 10.1016/j.brainres.2015.07.049 

The LDI corrects the proportion of correctly identified lures by accounting for the rate of “similar” responses to foils. This means that, when calculating the LDI, we consider how often participants tend to respond “similar” to items that were not studied. By doing so, we can distinguish between true lure discrimination and general familiarity that participants may have with the items in the task. This helps control for the familiarity effect, which could bias the results if not considered. This correction provides a more robust measure of discrimination.

- Section 3, behavioral results, currently reads as a summary of analysis results, yet the link between these results and the study’s broader scientific conclusions is ambiguous. For instance, in Lines 302-306, the finding of “more positive activity in a central-anterior cluster of electrodes for old hits compared to new misses” is presented as an observation. However, the inference or conclusion drawn from this observation, along with how it aligns with the study’s objectives, is not clearly articulated. 

The interpretation of the result mentioned, specifically regarding the observed “more positive activity in a central-anterior cluster of electrodes for old hits compared to new misses”, is detailed in the discussion section, which we have copied below for clarity.

Briefly, the more positive activity for old hits compared to new misses suggests a stronger memory trace for those categories that were subsequently recognized (old hit). In contrast, new miss, which implies unrecognition of the category, reflects a weaker trace. These findings align with previous SME studies, in which more positive ERPs during encoding are associated with subsequent successful recognition of those items. Thus, supporting our interpretation that enhanced ERP responses reflect strengthened memory encoding.

“ERPs from the encoding of image categories (when averaging all items) [see 31] that were correctly identified as old (old hits) were equivalent to the ERPs for those categories that were subsequently marked as similar (old false alarms indicates recognition of the category) and for those categories that were lures which attracted old responses (similar false alarms). Furthermore, ERPs from the encoding of image categories that were subsequently recognized (old hit) were significantly different from those categories that were subsequently marked as new (new miss indicates both category and item non-recognition). These results suggest a category recollection in which the consecutive presentation of similar items within a category created a strong category-related memory trace [20,31]. This trace is reflected in an increased ERP positivity of those categories which were previously seen and recognized in comparison to non-recognized and marked-as-new items, as the last one implied less memory encoding strengthening and consequently a weaker trace. These results are consistent with global matching models, which propose that the memory strength of a tested item arises from the similarity between its representation and all other representations from studied items (known as global similarity) [13]. Thus, higher neural global similarity during encoding leads to an increase in recognition memory [7-8,13]. However, strengthening of categories could also lead to an increased ERP positivity of similar false alarms, as the items were new but belonged to previously seen categories. In this way, old hit and similar FA had a more positive ERP than the new miss (Figure 5), although only the old hits showed differences at a statistical level.”